# Neuroprotective effects of TRPA1 channels in the cerebral endothelium following ischemic stroke

Paulo Wagner Pires, Scott Earley*

Department of Pharmacology, Center for Cardiovascular Research, University of Nevada, Reno, United States

**Abstract** Hypoxia and ischemia are linked to oxidative stress, which can activate the oxidant-sensitive transient receptor potential ankyrin 1 (TRPA1) channel in cerebral artery endothelial cells, leading to vasodilation. We hypothesized that TRPA1 channels in endothelial cells are activated by hypoxia-derived reactive oxygen species, leading to cerebral artery dilation and reduced ischemic damage. Using isolated cerebral arteries expressing a $Ca^{2+}$ biosensor in endothelial cells, we show that 4-hydroxynonenal and hypoxia increased TRPA1 activity, detected as TRPA1 sparklets. TRPA1 activity during hypoxia was blocked by antioxidants and by TRPA1 antagonism. Hypoxia caused dilation of cerebral arteries, which was disrupted by antioxidants, TRPA1 blockade and by endothelial cell-specific *Trpa1* deletion (*Trpa1* ecKO mice). Loss of TRPA1 channels in endothelial cells increased cerebral infarcts, whereas TRPA1 activation with cinnamaldehyde reduced infarct in wildtype, but not *Trpa1* ecKO, mice. These data suggest that endothelial TRPA1 channels are sensors of hypoxia leading to vasodilation, thereby reducing ischemic damage.
DOI: https://doi.org/10.7554/eLife.35316.001

## Introduction

Interruption of regional blood flow within the brain can rapidly cause irreparable neuronal damage. The cerebral circulation exhibits unique capabilities that allows it to adjust to varying environmental conditions and pathophysiological situations so as to maintain optimal perfusion and minimize such injury. G-protein-coupled receptors and ion channels present on the endothelial cells and vascular smooth muscle cells (SMCs) that form the walls of cerebral blood vessels initiate many of the signaling cascades that enable these intrinsic adaptive processes. The specific cellular pathways responsible for cerebrovascular homeostasis are of considerable interest as potential therapeutic targets for diseases associated with impaired blood flow regulation within the brain, such as stroke and vascular cognitive impairment; however, these pathways remain incompletely understood. We recently reported that transient receptor potential ankyrin 1 (TRPA1) cation channels are present in the endothelium of arteries within the brain, but not in other arterial beds (*Sullivan et al., 2015*), suggesting a specialized role for these channels in cerebral blood flow regulation. In the current study, we investigated the endogenous adaptive and protective functions of TRPA1 channels within the cerebral vasculature.

TRPA1, the sole member of the mammalian ankyrin TRP subfamily, is a large-conductance, $Ca^{2+}$-permeable, non-selective cation channel (*Nagata et al., 2005*; *Earley and Brayden, 2015*; *Karashima et al., 2010*). TRPA1 channels are present on perivascular nerves surrounding small mesenteric arteries, and their stimulation causes vasodilation through release of C-protein gene-related peptide (*Bautista et al., 2005*). In addition, TRPA1 channels form a $Ca^{2+}$-signaling complex with intermediate-conductance, $Ca^{2+}$-activated potassium channels ($K_{Ca}3.1$) at sites of close contact between cerebral artery endothelial cells and underlying SMCs (*Earley et al., 2009*). Consequently,

*For correspondence:
searley@med.unr.edu

Competing interests: The authors declare that no competing interests exist.

**eLife digest** A stroke can cause long-lasting physical and mental disabilities in patients including loss of mobility, speech defects and confusion. Most strokes happen when the blood supply to part of the brain is cut off due to blood clots or clumps of fat blocking blood vessels called arteries. To prevent a blocked blood vessel causing a stroke, the network of blood vessels in the brain contains alternative routes to each area. The arteries in these alternative routes can widen to allow more blood to flow through them and avoid the blockage.

When the blood supply to part of the brain is cut off, the level of oxygen in that area decreases. This causes highly reactive molecules known collectively as free radicals to be produced, which can bind to other molecules in cells and stop them from working properly. A protein called TRPA1 is found in the cells that form the inner lining of blood vessels. When it is active, TRPA1 forms a channel that allows signals known as calcium ions to enter the cell, which ultimately leads to arteries in the brain becoming wider. A free radical known as 4-HNE binds to TRPA1, but it is not clear if this enables the channel to directly sense the levels of oxygen in the brain.

Pires and Earley studied TRPA1 channels in brain arteries from mice. The experiments found that decreasing the levels of oxygen in the arteries caused 4-HNE to accumulate and activate TRPA1, resulting in the blood vessels becoming wider. Chemicals that inhibit the production of free radicals blocked the activity of the TRPA1 channels. Mice that lacked TRPA1 were more likely to sustain damage to the brain during strokes than normal mice. Furthermore, injecting normal mice experiencing a stroke with a drug that activates TRPA1 reduced the amount of damage to the brain.

The findings of Pires and Earley suggest that TRPA1 plays an important role in protecting the brain during strokes and other conditions that reduce the brain's blood supply. Future studies will assess whether drugs that activate TRPA1 have the potential to help reduce long-term disabilities in human patients who have a stroke.

DOI: https://doi.org/10.7554/eLife.35316.002

activation of TRPA1 channels, for example with the electrophilic compound allyl isothiocyanate (AITC), derived from mustard oil, induces endothelium-dependent vasodilation of cerebral resistance arteries, an effect that is prevented by blockade of $K_{Ca}3.1$ channels (*Earley et al., 2009*). Endogenous regulation of TRPA1 in the cerebral endothelium has been linked to reactive oxygen species (ROS), particularly superoxide anions ($O_2^-$) generated by NADPH oxidase 2 (NOX2) (*Sullivan et al., 2015*). ROS-induced activation of TRPA1 has been shown to occur through a process that requires generation of lipid peroxidation products that directly activate the channel, such as 4-hydroxynonenal (4-HNE) (*Sullivan et al., 2015*; *Trevisani et al., 2007*). Increased generation of ROS and oxidative stress is a hallmark of vascular diseases, but the importance of TRPA1-mediated vasodilation for the regulation of blood flow in the brain under pathological conditions remains unknown.

Cerebral blood vessels have the inherent ability to dilate in response to hypoxia (*Kontos et al., 1978*). When hypoxia occurs within specific regions of the brain, this adaptation increases localized blood flow to the affected area, enhancing delivery of oxygen. Although the molecular and cellular mechanisms responsible for this important response are not known, a recent study suggested that TRPA1 channels are sensitive to changes in $O_2$ concentration and are activated by both hyperoxic and hypoxic conditions (*Takahashi et al., 2011*). In addition, hypoxia and ischemia are linked to elevated ROS production and increased generation of 4-HNE (*Schmidt et al., 1996*; *Kunstmann et al., 1996*), an endogenous activator of TRPA1 activity.

Here, we tested the hypothesis that hypoxia causes activation of TRPA1 channels in the cerebral endothelium, resulting in vasodilation, and further propose that this response constitutes an adaptive response during perfusion deficiency. Using an integrative approach and a newly developed mouse model expressing a $Ca^{2+}$ biosensor exclusively in endothelial cells, we elucidated the underlying molecular mechanisms responsible for activation of TRPA1 channels in the intact cerebral endothelium during hypoxia and investigated the pathophysiological impact of this novel pathway in vivo using an established model of ischemic strokes. Our findings suggest that TRPA1 channels in the cerebral endothelium are early sensors of hypoxia and initiate an adaptive response to reduce ischemic damage in the brain.

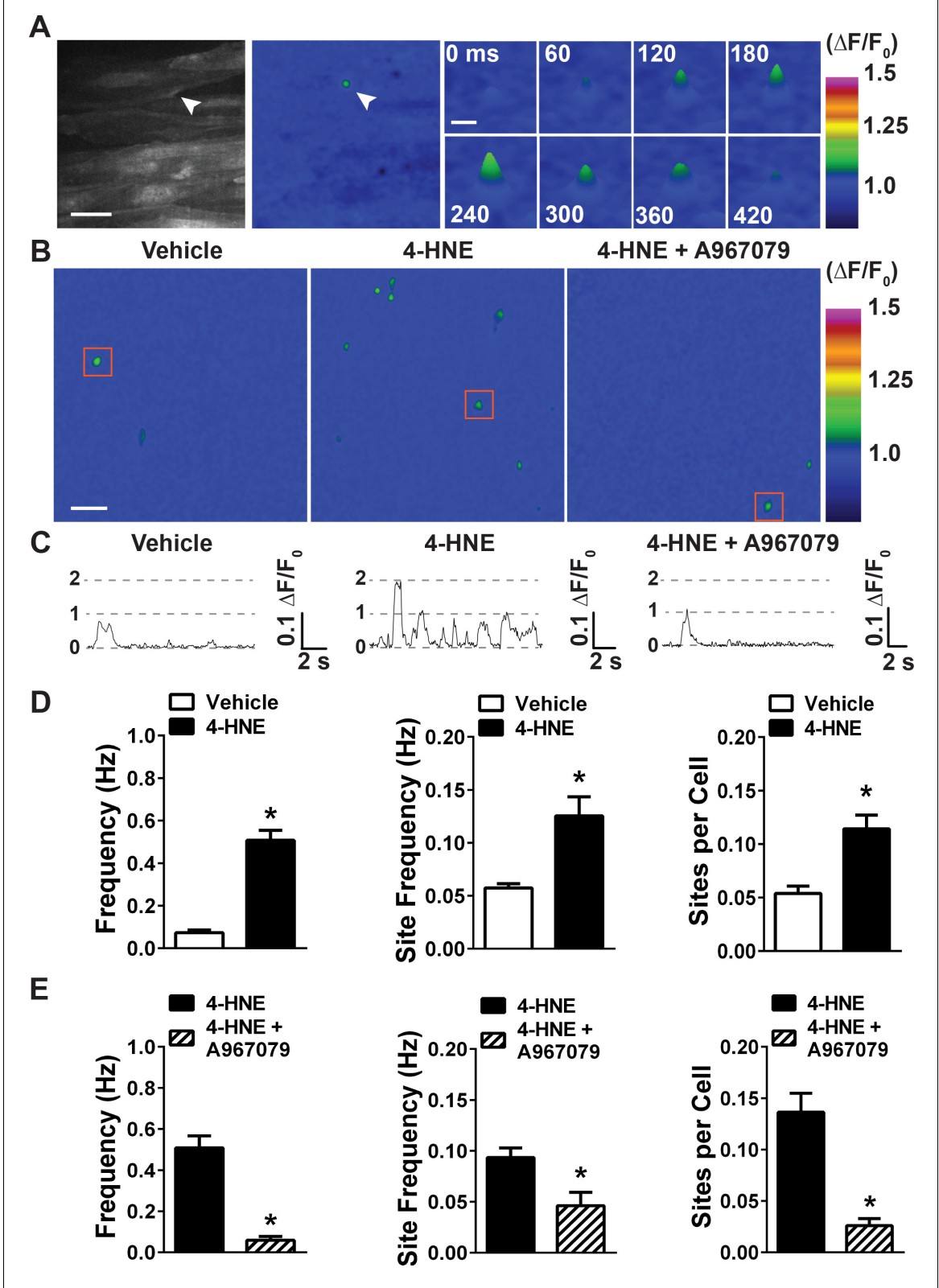

**Figure 1.** 4-HNE stimulates TRPA1 sparklets in the endothelium of intact cerebral arteries. (**A**) Representative images of endothelial cells from cerebral arteries from *Tek:Gcamp6f* mice mounted *en face*, presented in grayscale (left) and pseudocolored (middle and right panels). Scale bar = 15 μm. The images on the right are a timelapse and digital magnification of the TRPA1 sparklet indicated by the arrow in the right and middle panels. Scale bar = 5 μm. (**B**) Representative pseudocolored images of a 512 × 512 pixels field of view from *Tek:Gcamp6f* mice showing sparklets (green) after exposure to

*Figure 1 continued*

vehicle, 4-HNE and 4-HNE + A967079. Scale bar = 20 μm. (C) Representative $\Delta F/F_0$ vs. time plots for a single sparklet site. 0, 1, two levels indicate hypothesized numbers of TRPA1 channels engaged during each signal. (D) Summary data showing the effects of the TRPA1 channel activator 4-HNE (1 μM) on TRPA1 sparklet frequency, site frequency and number of sites per cell (n = 24 fields of view from four different arteries, N = 4 mice). (E) TRPA1 inhibition with A967079 (1 μM) significantly prevented 4-HNE induction of TRPA1 sparklets in endothelial cells (20 fields of view from five different arteries; N = 3 mice). Data are presented as means ± SEM (*p<0.05, *Student*'s t-test). TRPA1 sparklets were recorded in the presence of the cell permeable $Ca^{2+}$ chelator EGTA-AM (10 μM) and the sarcoendoplasmic reticulum $Ca^{2+}$-ATPase inhibitor cyclopiazonic acid (CPA, 30 μM).

DOI: https://doi.org/10.7554/eLife.35316.003

The following source data and figure supplements are available for figure 1:

**Source data 1.** Excel spreadsheet containing the individual numeric values of the parameters analyzed in *Figure 1*.
DOI: https://doi.org/10.7554/eLife.35316.010

**Figure supplement 1.** TRPA1 is present in the endothelium of cerebral arteries.
DOI: https://doi.org/10.7554/eLife.35316.004

**Figure supplement 2.** Reporter expression of cre-recombinase using the mT/mG reporter mice.
DOI: https://doi.org/10.7554/eLife.35316.005

**Figure supplement 3.** Extracellular $Ca^{2+}$ is required for TRPA1 sparklets.
DOI: https://doi.org/10.7554/eLife.35316.006

**Figure supplement 3—source data 1.** Excel spreadsheet containing the individual numeric values of the parameters analyzed in *Figure 1—figure supplement 2*.
DOI: https://doi.org/10.7554/eLife.35316.007

**Figure supplement 4.** Properties of 4-HNE-induced TRPA1 sparklets.
DOI: https://doi.org/10.7554/eLife.35316.008

**Figure supplement 4—source data 1.** Excel spreadsheet containing the individual numeric values of the parameters analyzed in *Figure 1—figure supplement 3*.
DOI: https://doi.org/10.7554/eLife.35316.009

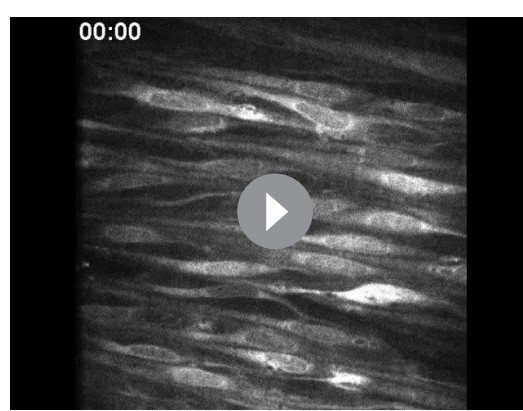

**Video 1.** Spontaneous $Ca^{2+}$ transients in the endothelium of a cerebral artery. Representative movie of a cerebral artery from a *Tek:Gcamp6f* mouse mounted *en face* and superfused with normoxic (21% $O_2$, 6% $CO_2$, 73% $N_2$), warm (37°C) physiological saline solution (PSS) without EGTA-AM or CPA.
DOI: https://doi.org/10.7554/eLife.35316.011

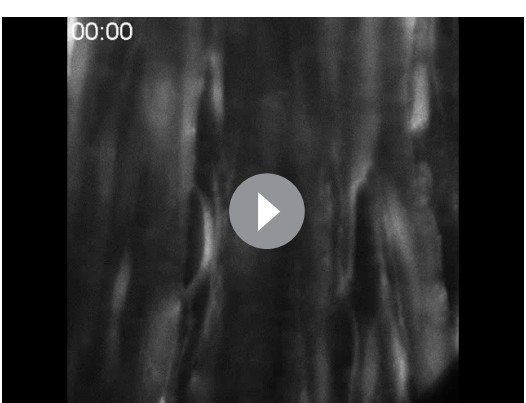

**Video 2.** $Ca^{2+}$ events in the endothelium of a mesenteric artery from a *Tek:Gcamp6f* mouse. Representative movie of a mesenteric artery from a *Tek:Gcamp6f* mouse mounted *en face* and superfused with normoxic (21% $O_2$, 6% $CO_2$, 73% $N_2$), warm (37°C) physiological saline solution (PSS) without EGTA-AM or CPA.
DOI: https://doi.org/10.7554/eLife.35316.012

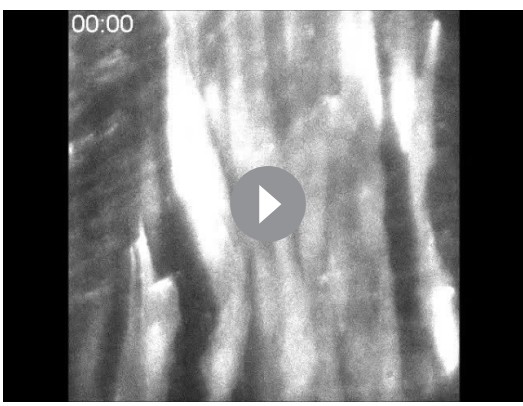

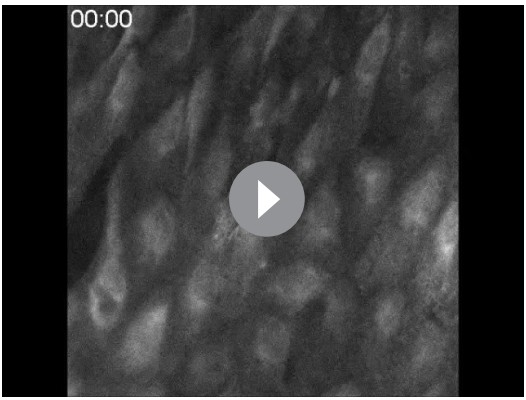

**Video 3.** Ca$^{2+}$ events in the endothelium of a skeletal muscle artery from a *Tek:Gcamp6f* mouse. Representative movie of an skeletal muscle artery from a *Tek:Gcamp6f* mouse mounted *en face* and superfused with normoxic (21% O$_2$, 6% CO$_2$, 73% N$_2$), warm (37°C) physiological saline solution (PSS) without EGTA-AM or CPA.
DOI: https://doi.org/10.7554/eLife.35316.013

**Video 4.** Ca$^{2+}$ events in the pulmonary endothelium from a *Tek:Gcamp6f* mouse. Representative movie of a pulmonary artery from a *Tek:Gcamp6f* mouse mounted *en face* and superfused with normoxic (21% O$_2$, 6% CO$_2$, 73% N$_2$), warm (37°C) physiological saline solution (PSS) without EGTA-AM or CPA.
DOI: https://doi.org/10.7554/eLife.35316.014

## Results

### New genetically encoded Ca$^{2+}$ biosensor mice enable optical recording of single-channel TRPA1 activity in the endothelium of intact cerebral arteries

A primary goal of this study was to determine the direct effects of hypoxia on TRPA1 channels that are present in the endothelium of intact cerebral arteries (*Figure 1—figure supplement 1*). An effective strategy for directly measuring changes in channel activity is to record local, transient elevations in cytosolic Ca$^{2+}$ levels generated by Ca$^{2+}$ influx through single TRPA1 channels in the endothelium, detected as TRPA1 sparklets (*Sullivan et al., 2015*), using optical patch-clamp methods (*Sullivan and Earley, 2013*) (*Figure 1A*). A previous study reported the use of transgenic mice expressing the *Gcamp2* Ca$^{2+}$ biosensor under the control of the Cx40 promoter to record TRPV4 sparklets in the endothelium of mesenteric arteries (*Sonkusare et al., 2012*). However, we found that the fluorescence intensity of Cx40-based *Gcamp2* (and *Gcamp5*) Ca$^{2+}$ biosensors was very low in the endothelium of cerebral arteries, preventing reliable recordings from being obtained. We also found that the cerebral endothelium could not be effectively loaded with standard Ca$^{2+}$ indicator dyes, such as Fluo-4AM. To overcome these limitations, we generated a new transgenic mouse line that expresses *Gcamp6f*, a Ca$^{2+}$ biosensor with fast kinetics (*Chen et al., 2013*), exclusively in the endothelium. To accomplish this, we crossed mice heterozygous for the expression of a floxed *Stop* codon upstream of the *Gcamp6f* gene (see Materials and methods) with mice heterozygous for the expression of *cre*-recombinase under the control of the endothelial-specific *Tek* promoter/enhancer (*Tek$^{cre}$*). The *mT/mG* reporter mouse (*Muzumdar et al., 2007*) was used to confirm that *cre*-recombinase is only

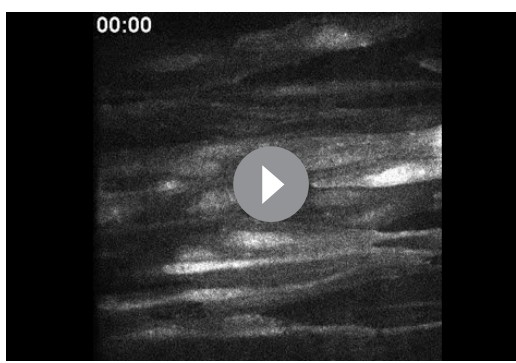

**Video 5.** Spontaneous Ca$^{2+}$ influx events the cerebral endothelium. Representative movie of a cerebral artery isolated from a *Tek:Gcamp6f* mouse mounted *en face* and superfused with normoxic (21% O$_2$, 6% CO$_2$, 73% N$_2$), warm (37°C) physiological saline solution (PSS) in the presence of EGTA-AM (10 µM) and CPA (30 µM).
DOI: https://doi.org/10.7554/eLife.35316.015

expressed in the endothelium of cerebral arteries from *Tek^cre* mice (**Figure 1—figure supplement 2**). We found that cerebral, mesenteric, skeletal muscle and pulmonary arteries isolated from *Tek: Gcamp6f* mice expressed the $Ca^{2+}$ biosensor only in the endothelium and provided excellent signal-to-noise ratio for optical detection of spontaneous and evoked $Ca^{2+}$ signals in this tissue (**Videos 1–4**).

To initially characterize TRPA1 sparklets in the intact cerebral endothelium, we mounted pial arteries from *Tek:Gcamp6f* mice *en face* as previously described (**Sonkusare et al., 2012**). The endothelium of arteries mounted *en face* is not subjected to physiological levels of luminal shear stress and intraluminal pressure and cannot be used to study how these stimuli affect endothelial cell function. However, Sonkusare *et al.* showed that the TRPV4 sparklets recorded from the endothelium of arteries mounted *en face* are statistically indistinguishable from those recorded from intact arteries pressurized at physiological levels, suggesting that longitudinal incision of the artery does not alter basic ion channel properties (**Sonkusare et al., 2012**). To isolate $Ca^{2+}$ influx events, we treated initial preparations with the sarcoplasmic/endoplasmic reticulum $Ca^{2+}$-ATPase (SERCA) inhibitor cyclopiazonic acid (CPA, 30 µM) to prevent release of $Ca^{2+}$ from intracellular stores. Vessels used for these $Ca^{2+}$ signaling experiments were also treated with the cell-permeant $Ca^{2+}$ chelator EGTA-AM (10 µM) to limit the intracellular diffusion of $Ca^{2+}$ influx and improve the signal-to-noise ratio.

Using these conditions, the endothelium was exposed to a near-maximally effective concentration (1 µM) of the endogenous TRPA1 agonist 4-HNE (**Sullivan et al., 2015**) via the superfusing bath. $Ca^{2+}$ events were recorded at an average of ~40 frames/s in a 512 × 512 pixel field of view (pixel size = 0.27 µm) containing approximately 40 endothelial cells (mean area of cerebral artery endothelial cells = 462 ± 24 $µm^2$, n = 25 cells, N = 5 mice). We observed that 4-HNE significantly increased the frequency of highly localized, transient $Ca^{2+}$ signals by approximately 10-fold compared with vehicle controls (0.51 ± 0.05 Hz vs. 0.07 ± 0.01 Hz) (**Figure 1D** and **Videos 5** and **6**). Notably, these signals exhibited distinct amplitude levels and duration reminiscent of single-channel activity recorded using patch-clamp electrophysiology (**Demuro and Parker, 2004**) (**Figure 1C**). The selective TRPA1 antagonist A967079 (1 µM) (**Chen et al., 2011**) significantly diminished the 4-HNE–induced increase in activity, reducing the frequency of these $Ca^{2+}$ signals from 0.51 ± 0.06 Hz to 0.06 ± 0.02 Hz (**Figure 1E**). The frequency of $Ca^{2+}$ signals induced by exposure to 4-HNE was also significantly reduced by removal of extracellular $Ca^{2+}$, confirming that these signals are generated by an influx of $Ca^{2+}$ (**Figure 1—figure supplement 3**). Collectively, these data identify the local, transient $Ca^{2+}$ signals evoked by administration of 4-HNE as *bona fide* TRPA1 sparklets.

Further analyses showed that the number of active sites per field of view significantly increased from 1.68 ± 0.27 under control conditions to 4.41 ± 0.52 in the presence of 4-HNE (**Figure 1B** and

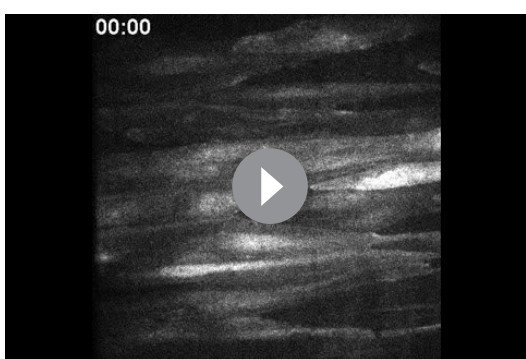

**Video 6.** 4-HNE stimulates TRPA1 sparklets the cerebral artery endothelium. Representative movie of the same field of view as in **Video 5** showing that application of 4-HNE (1 µM) significantly increased the frequency of TRPA1 sparklets in the cerebral endothelium. TRPA1 sparklets were recorded in the presence of EGTA-AM (10 µM) and CPA (30 µM).
DOI: https://doi.org/10.7554/eLife.35316.016

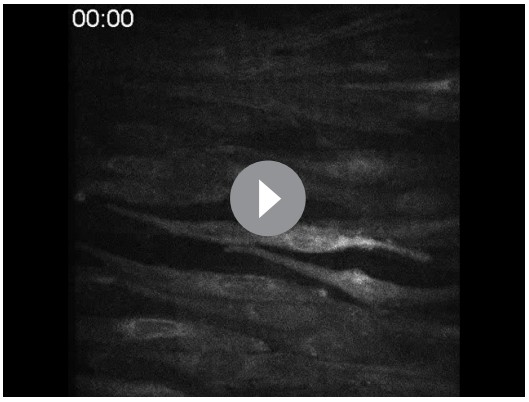

**Video 7.** Spontaneous large $Ca^{2+}$transients in the endothelium of a cerebral artery. Representative movie of a cerebral artery from a *Tek:Gcamp6f* mouse mounted *en face* and superfused with normoxic (21% $O_2$, 6% $CO_2$, 73% $N_2$), warm (37°C) physiological saline solution (PSS) without EGTA-AM or CPA.
DOI: https://doi.org/10.7554/eLife.35316.019

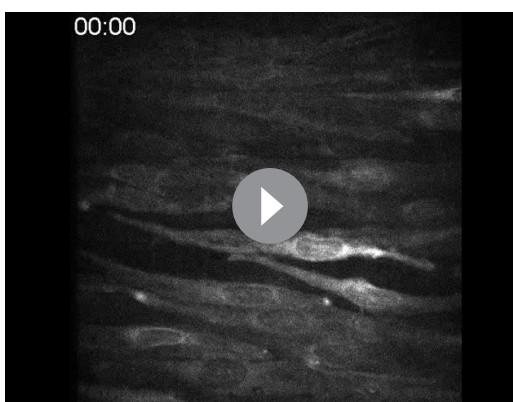

**Video 8.** 4-HNE increases the frequency of $Ca^{2+}$ transients in the cerebral artery endothelium. Representative movie of the same field of view as in *Video 7* showing that 4-HNE (1 μM) greatly increases the frequency of $Ca^{2+}$ transients in the endothelium. Events were recorded in the absence of EGTA-AM or CPA (30 μM).
DOI: https://doi.org/10.7554/eLife.35316.020

*D*, data shown as number of sites per cell). In addition, sparklet frequency at existing sites significantly increased from 0.06 ± 0.01 Hz at baseline to 0.13 ± 0.02 Hz following application of 4-HNE (*Figure 1C and D*). These findings indicate that the increase in TRPA1 sparklet frequency induced by administration of 4-HNE resulted from both the recruitment of previously inactive sparklet sites as well as an increase in the frequency of previously active sites. TRPA1 inhibition with the selective inhibitor A967079 largely prevented the increases in the number of active sparklets sites as well as the increase in sparklet frequency at previously active sites (*Figure 1B,C and E*). A plot of the amplitudes ($\Delta F/F_0$) of individual TRPA1 sparklets revealed a Gaussian distribution with a mode of 1.10 (*Figure 1—figure supplement 4*). The mean attack time (half-time to reach peak signal amplitude), decay time (half-time from peak to loss of signal), and duration of mode-amplitude TRPA1 sparklets were 87 ± 7 ms, 84 ± 6 ms, and 373 ± 23 ms, respectively. The mean spatial spread of TRPA1 sparklets was approximately 17.2 ± 0.6 μm$^2$, or ~3% of the total surface area of a single endothelial cell in this intact vascular preparation (*Supplementary file 1*).

## Stimulation of TRPA1 with 4-HNE also increases the frequency of larger $Ca^{2+}$ transients

We previously reported that stimulation of TRPA1 channels with the electrophilic compound allyl isothiocyanate (AITC) increased the frequency of dynamic $Ca^{2+}$ release from the ER due to $Ca^{2+}$-induced $Ca^{2+}$ release (*Qian et al., 2013*). To determine if 4-HNE -stimulated TRPA1 sparklets have a similar effect on ER $Ca^{2+}$ release, we exposed cerebral arteries from *Tek:Gcamp6f* mice mounted *en face* to 4-HNE in the absence of CPA or EGTA-AM. Under these conditions, the frequency of spontaneous transient events was significantly higher compared with arteries treated with CPA and EGTA-AM. 4-HNE significantly increased the frequency of $Ca^{2+}$ transients (0.76 ± 0.08 vs. 2.44 ± 0.20 Hz, *Videos 7* and *8*, *Figure 2C*), and this response was blocked by the TRPA1 channel inhibitor A967079 (1.12 ± 0.09 Hz, *Figure 2C* and *Video 9*). Spontaneous and 4-HNE-evoked $Ca^{2+}$ transients were not significantly different in amplitude, but were larger in terms of spatial spread and mean duration compared with TRPA1 sparklets (*Supplementary file 1* and *2*). The increase in $Ca^{2+}$ transients appeared to be a consequence of recruitment of previously inactive sites (number of sites per cell: 0.26 ± 0.02 vs. 1.44 ± 0.24), as well as an increase in frequency of $Ca^{2+}$ release from previously active sites (0.072 ± 0.004 vs. 0.121 ± 0.005 Hz, *Figure 2A–C*). TRPA1 inhibition with A967079 diminished the recruitment of new sites of $Ca^{2+}$ transients (number of sites per cell: 0.29 ± 0.02, *Figure 2A and C*) and reduced the frequency of $Ca^{2+}$ transients in previously active sites (0.098 ± 0.003 Hz, *Figure 2B and C*). These data suggest that TRPA1 sparklets induce $Ca^{2+}$ release from the ER, thereby amplifying the initial $Ca^{2+}$ influx signal.

## Hypoxia induces accumulation of 4-HNE and increases TRPA1 sparklet frequency in the endothelium of intact cerebral arteries

To test the hypothesis that hypoxia induces production of 4-HNE in the endothelium, *en face* cerebral arteries were incubated in normoxic or hypoxic conditions. Hypoxia was achieved by superfusing the tissue with physiological saline solution (PSS) equilibrated with a hypoxic gas mixture (5% $O_2$, 6% $CO_2$, 89% $N_2$). The $pO_2$ measured in the recording bath under these conditions was 13 ± 2 mmHg (n = 5). The pH of the superfusing solution was monitored in real time, and was constantly maintained between 7.37 and 7.42 during normoxic and hypoxic conditions. We observed an increase in

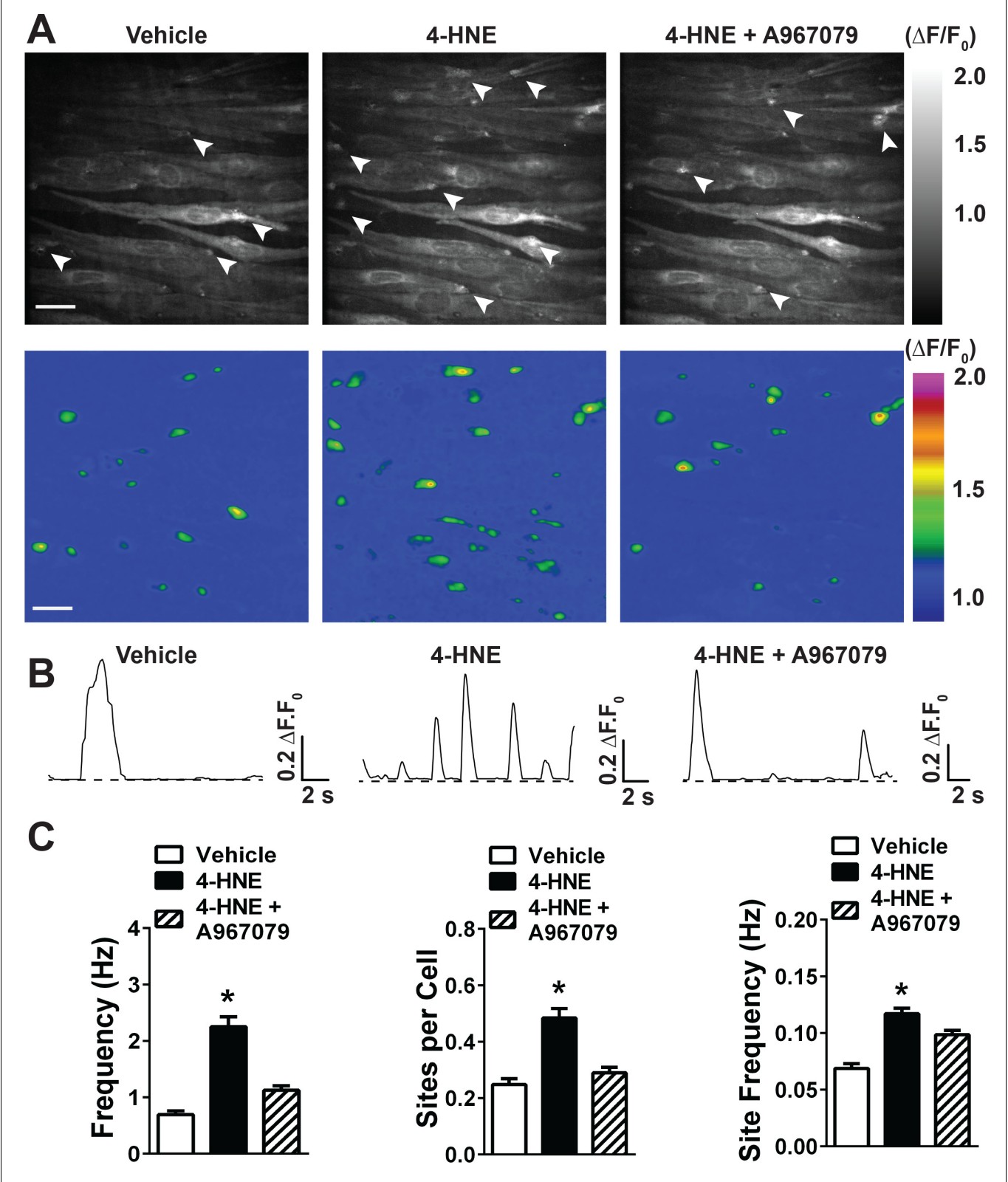

**Figure 2.** 4-HNE stimulates large $Ca^{2+}$transients in the endothelium of intact cerebral arteries. (**A**) Representative grayscale images of endothelial cells from *Tek:Gcamp6f* mice mounted *en face* in the absence of CPA and EGTA-AM. The white arrowheads point to sites of $Ca^{2+}$ transients. Scale bar = 30 μm. Pseudocolored representations of the images are shown below. Scale bar = 30 μm. (**B**) Representative $\Delta F/F_0$ vs. time plots of $Ca^{2+}$ transients from a single site of $Ca^{2+}$ transients. (**C**) Summary graphs showing that 4-HNE (1 μM) significantly increases frequency, number of active sites, and site

*Figure 2 continued on next page*

*Figure 2 continued*

frequency of Ca$^{2+}$ transients in cerebral artery endothelial cells. This response was diminished by the TRPA1 blocker A967079 (1 μM). (25 fields of view from three different arteries; N = 3 mice). Data are presented as means ± SEM (*p<0.05, one-way ANOVA).

DOI: https://doi.org/10.7554/eLife.35316.017

The following source data is available for figure 2:

**Source data 1.** Excel spreadsheet containing the individual numeric values of the parameters analyzed in *Figure 2*.

DOI: https://doi.org/10.7554/eLife.35316.018

4-HNE immunolabeling in cerebral arteries superfused with hypoxic PSS when compared to arteries superfused with normoxic PSS (*Figure 3A* and *Figure 3—figure supplement 1*).

Arteries from *Tek:Gcamp6f* mice were used to test the hypothesis that acute hypoxic exposure acts through TRPA1 channels to increase the frequency of Ca$^{2+}$ influx events in the cerebral endothelium. Acute hypoxic exposure significantly increased the frequency of TRPA1 sparklets, a response that was significantly reduced by A967079 (*Figure 3B–3D* and *Videos 10* and *11*). The increase in TRPA1 sparklet frequency induced by hypoxia reflected an increase in the number of active sparklets sites (*Figure 3B and D*) as well as the frequency of sparklets at active sites (control, 0.049 ± 0.007 Hz; hypoxia, 0.089 ± 0.007 Hz; p<0.05, *Figure 3C and D*); both increases were prevented by the TRPA1 inhibitor A967079, which restored TRPA1 sparklet frequency to levels that were not significantly different from normoxic conditions (0.057 ± 0.007 Hz, *Figure 3B* – D). The amplitude, kinetics, and spatial spread of hypoxia-evoked TRPA1 sparklets (*Supplementary file 1* and *Figure 3—figure supplement 2*) did not significantly differ from those of 4-HNE–induced TRPA1 sparklets. These data demonstrate that pathophysiologically relevant levels of hypoxia acutely stimulate TRPA1 activity in the endothelium of intact cerebral arteries.

## Hypoxia-induced increases in mitochondrial ROS stimulate TRPA1 activity in the cerebral endothelium

We next investigated the molecular mechanisms responsible for activation of TRPA1 channels in endothelial cells during hypoxia. Our prior study demonstrated that extracellular O$_2^-$ generated by NOX2 stimulates the formation of 4-HNE, which in turn activates TRPA1 channels in the cerebral endothelium (*Sullivan et al., 2015*). Unexpectedly, we found that extracellular superoxide dismutase (SOD) and the NOX inhibitor apocynin did not significantly inhibit hypoxia-induced increases in TRPA1 sparklet frequency (*Figure 4—figure supplement 1*), suggesting the involvement of an alternative pathway. Consistent with this latter possibility, we found that hypoxia-induced increases in TRPA1 sparklet frequency and sites per cell were significantly reduced by membrane-permeant PEG-SOD, suggesting that intracellular generation of ROS is essential for this response (*Figure 4B*). This conclusion is supported by imaging experiments showing that hypoxia caused a time-dependent increase in the fluorescence intensity of the ROS indicator dihydroethidium (DHE) in freshly isolated cerebral artery endothelial cells, which was prevented by PEG-SOD (*Figure 4A*).

Having eliminated NOX as a potential source of increased ROS production during hypoxia, we investigated the involvement of other pathways. A previous study suggested that generation of ROS by mitochondrial respiration is increased during hypoxia (*Hernansanz-Agustín et al., 2014*). In agreement with this report, we found

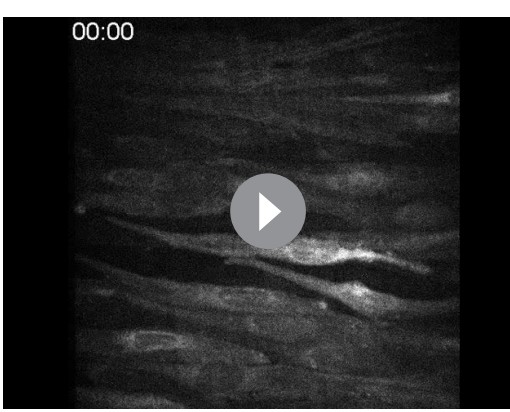

**Video 9.** TRPA1 channels mediate the increase in Ca$^{2+}$transients caused by 4-HNE. Representative movie of the same field of view as in *Videos 7* and *8* showing that TRPA1 inhibition with A967079 (1 μM) prevents the increase in Ca$^{2+}$ transients frequency elicited by 4-HNE (1 μM) in the endothelium. TRPA1 sparklets were recorded in the presence of EGTA-AM (10 μM) and CPA (30 μM).

DOI: https://doi.org/10.7554/eLife.35316.021

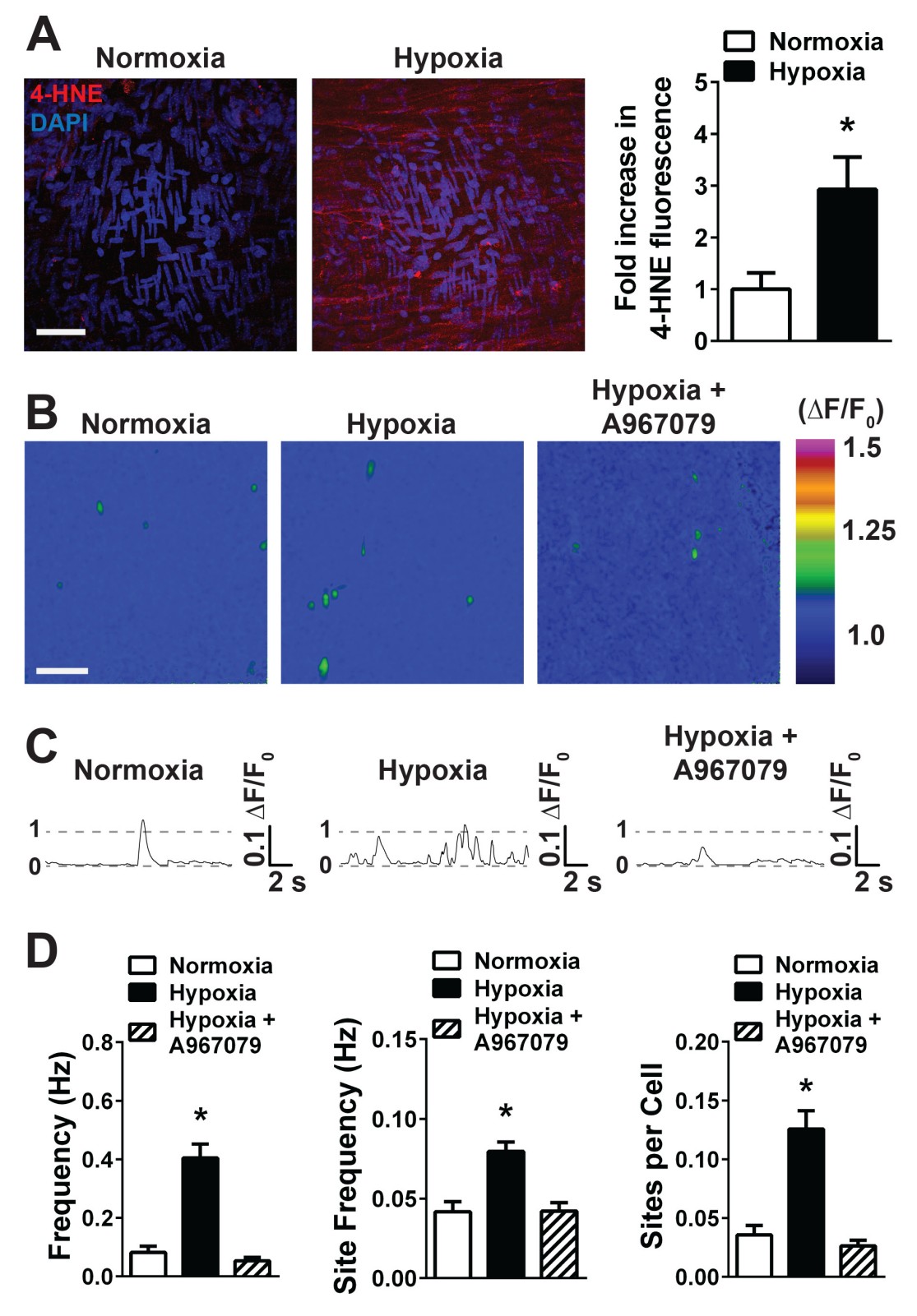

**Figure 3.** Acute hypoxia increases 4-HNE accumulation and increases TRPA1 sparklet frequency in the cerebral endothelium. (**A**) Representative maximum intensity projection of Z-stacks of cerebral arteries mounted *en face and* exposed to PSS equilibrated with a normoxic (21% $O_2$, 6% $CO_2$, 73% $N_2$, left panel) or hypoxic gas mixture (5% $O_2$, 6% $CO_2$, 89% $N_2$, right panel) and immunolabeled for 4-HNE (red). Scale bar = 40 µm, nuclei of cells are labeled by DAPI (blue). Acute hypoxia significantly increased 4-HNE immunoreactivity in cerebral arteries (*p<0.05 *Student*'s t-test, N = 10–9 fields of

*Figure 3 continued on next page*

*Figure 3 continued*

view from three different experiments). (B) Representative pseudocolored images of cerebral arteries from *Tek:Gcamp6f* mice mounted *en face* and exposed to normoxic (left panel) or hypoxic (middle and right panels) PSS in the presence or absence of the selective TRPA1 blocker A967079 (1 µM). Green: active TRPA1 sparklet sites. Scale bar = 20 µm. (C) Representative $\Delta F/F_0$ vs. time plots for a single sparklet site showing an increase in TRPA1 sparklet frequency during hypoxia which was significantly inhibited by the TRPA1 blocker A967079. (D) Summary data showing the effects of hypoxia on TRPA1 sparklet frequency (left), site frequency (middle) and number of sites per cell (right) in the presence and absence of the selective TRPA1 inhibitor A967079 (1 µM). (*$p < 0.05$, one-way ANOVA; N = 25–28 – 33 fields of view from six different arteries isolated from six mice). TRPA1 sparklets were recorded in the presence EGTA-AM (10 µM) and CPA (30 µM).

DOI: https://doi.org/10.7554/eLife.35316.022

The following source data and figure supplements are available for figure 3:

**Source data 1.** Excel spreadsheet containing the individual numeric values of the parameters analyzed in *Figure 3*.

DOI: https://doi.org/10.7554/eLife.35316.027

**Figure supplement 1.** Hypoxia causes 4-HNE accumulation in cerebral arteries.

DOI: https://doi.org/10.7554/eLife.35316.023

**Figure supplement 1—source data 1.** Excel spreadsheet containing the individual numeric values of the parameters analyzed in *Figure 3—figure supplement 1*.

DOI: https://doi.org/10.7554/eLife.35316.024

**Figure supplement 2.** Properties of hypoxia-induced TRPA1 sparklets.

DOI: https://doi.org/10.7554/eLife.35316.025

**Figure supplement 2—source data 1.** Excel spreadsheet containing the individual numeric values of the parameters analyzed in *Figure 3—figure supplement 2*.

DOI: https://doi.org/10.7554/eLife.35316.026

that pre-incubation of freshly isolated endothelial cells with the mitochondrial-targeted antioxidant mitoTEMPO (500 nM) significantly inhibited hypoxia-induced increases in DHE fluorescence (*Figure 4A*), suggesting that mitochondria are the primary source of intracellular ROS generated during hypoxia. To determine if mitochondrial ROS are responsible for the activation of TRPA1 channels during hypoxia, we treated cerebral arteries from *Tek:Gcamp6f* mice with mitoTEMPO prior to hypoxia exposure. This treatment significantly diminished hypoxia-induced increases in TRPA1 sparklet frequency, indicating a critical role for mitochondrial ROS generation in this response (*Figure 4C*).

## Hypoxia dilates cerebral pial arteries and penetrating arterioles by activating TRPA1 channels in the endothelium

Pressure myography studies using intact cerebral pial arteries were carried out to study the influence of TRPA1 channels on hypoxia-induced dilation. Arteries were pressurized to physiological levels (60 mmHg) and allowed to generate spontaneous myogenic tone. Acute exposure to hypoxia ($pO_2$ = 13 ± 2 mmHg) induced vasodilation that was significantly diminished by disruption of endothelial cell function by passing an air bubble through the vascular lumen (*Ralevic et al., 1989*) (*Figure 5A*), indicating involvement of the endothelium. Vasodilation in response to hypoxia was also inhibited by Tempol (100 µM) (*Figure 5B*), but was not significantly reduced by NOX2 inhibition and extracellular SOD (*Figure 5—figure supplement 1*), supporting the concept that intracellular ROS generation is required for this response. Further, we found that hypoxia-induced dilation was significantly blunted by mitoTEMPO (500 nM) (*Figure 5C*), suggesting that mitochondria are the source of ROS required

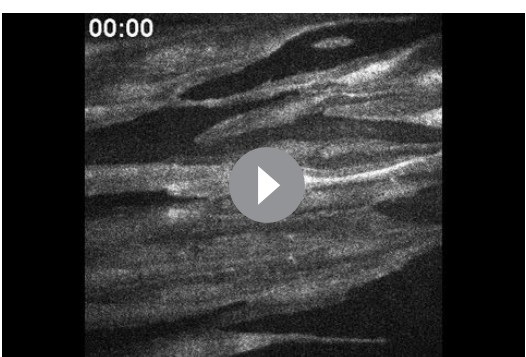

**Video 10.** Spontaneous $Ca^{2+}$ influx events the cerebral endothelium during normoxia. Representative movie of a cerebral artery isolated from a *Tek:Gcamp6f* mouse mounted *en face* and superfused with normoxic (21% $O_2$, 6% $CO_2$, 73% $N_2$), warm (37°C) physiological saline solution (PSS) in the presence of EGTA-AM (10 µM) and CPA (30 µM).

DOI: https://doi.org/10.7554/eLife.35316.028

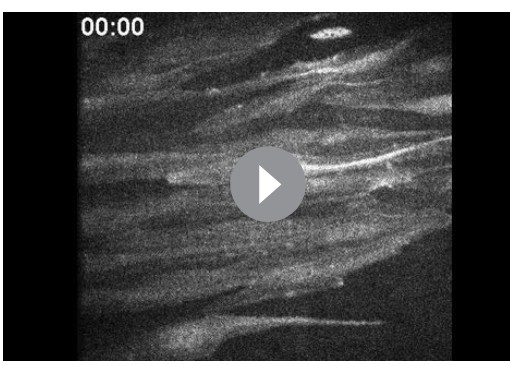

**Video 11.** Hypoxia stimulates TRPA1 sparklet frequency in the cerebral endothelium. Representative movie of the same field of view from **Video 10** showing that superfusing the preparation with hypoxic (5% $O_2$, 6% $CO_2$, 89% $N_2$), warm (37°C) physiological saline solution (PSS) increases the frequency and number of active sites of TRPA1 sparklets. TRPA1 sparklets were recorded in the presence of EGTA-AM (10 µM) and CPA (30 µM).

DOI: https://doi.org/10.7554/eLife.35316.029

for this response. Control experiments showed that Tempol and mitoTEMPO had no significant effect on cerebral artery dilation in response to administration of 4-HNE, indicating that these drugs do not directly inhibit this TRPA1-dependent response (**Figure 5—figure supplement 2**).

Vasodilation in response to hypoxic exposure was significantly reduced by blockade of TRPA1 channels with A967079 (**Figure 6A**) and was diminished in cerebral arteries isolated from endothelial cell-specific TRPA1 knockout mice (*Trpa1* ecKO) (**Sullivan et al., 2015**) compared with TRPA1 floxed, cre-recombinase negative, littermate controls (*Trpa1*<sup>fl/fl</sup>, **Figure 6B**). To investigate potential vasodilator mechanisms acting downstream of TRPA1 channels, we blocked $K_{Ca}3.1$ channels with TRAM34 and $K_{Ca}2.3$ channels with apamin. This treatment significantly reduced hypoxia-induced dilation (**Figure 6C**). Together, these data demonstrate that hypoxia-induced increases in mitochondrial ROS production stimulate TRPA1-mediated $Ca^{2+}$ influx in the cerebral artery endothelium, causing vasodilation of pial arteries through activation of $K_{Ca}3.1$ and/

or $K_{Ca}2.3$ channels.

We also investigated the effects of TRPA1 activity on vasomotor activity of cerebral penetrating arterioles. Brain slices (200 µm in thickness) from perfusion-fixed *Tek*<sup>gfp</sup> mice (Tg(TIE2GFP)287Sato/J) reporter mice that express GFP in the endothelium (**Motoike et al., 2000**) were immunolabeled for TRPA1. Penetrating arterioles were identified as branches from the surface pial arteries that entered the underlying brain parenchyma. TRPA1 immunofluorescence was detected in the endothelium of penetrating arterioles (**Figure 7A**, left panels, red) but was not detected when the primary antibody for TRPA1 was omitted, although GFP fluorescence was observed (**Figure 7A**, right panels). Functional ex vivo pressure myography experiments showed that exposing penetrating arterioles to 4-HNE induced dilation that was significantly diminished by A967079 (**Figure 7B**). In addition, exposure of pressurized penetrating arterioles to hypoxia caused dilation that was blunted by A967079 (**Figure 7C**). Together, these data suggest that TRPA1 channels are present and functional in penetrating arterioles and elicit vasodilation in response to 4-HNE and hypoxia.

## Endothelial cell TRPA1 activity decreases damage following ischemic stroke

Our data indicate that hypoxia-induced activation of TRPA1 channels in the endothelium of cerebral arteries and penetrating arterioles causes vasodilation. We propose that hypoxia-induced vasodilation of cerebral arteries is an adaptive response that serves to improve collateral perfusion within affected brain regions following ischemic stroke. To test this hypothesis, we induced ischemic strokes in *Trpa1* ecKO mice and *TRPA1*<sup>fl/fl</sup> control littermates using the middle cerebral artery occlusion (MCAO) model (**Longa et al., 1989**) without reperfusion. We found that although reduction in cerebral perfusion after MCAO did not significantly differ between *Trpa1*<sup>fl/fl</sup> and *Trpa1* ecKO mice (**Figure 8—figure supplement 1**), infarct size was significantly increased in *Trpa1* ecKO mice compared with littermate controls 24 hr after MCAO (**Figure 8A**). In companion interventional studies, we examined the effects of stimulating TRPA1 activity with cinnamaldehyde (CinA) after MCAO. Using pressure myography, we found that CinA induced a concentration-dependent dilation of pressurized cerebral arteries isolated from control mice, a response that was absent in cerebral arteries isolated from *Trpa1* ecKO mice (**Figure 8—figure supplement 2A and B**). CinA (30 µM) also induced dilation of penetrating arterioles, a response that was significantly blunted by A967079 (**Figure 8—figure supplement 2C and D**). Pharmacological activation of TRPA1 channels in vivo by intraperitoneal (i.p.) injection of CinA (50 mg/kg) (**Huang et al., 2007**), administered 15 min after

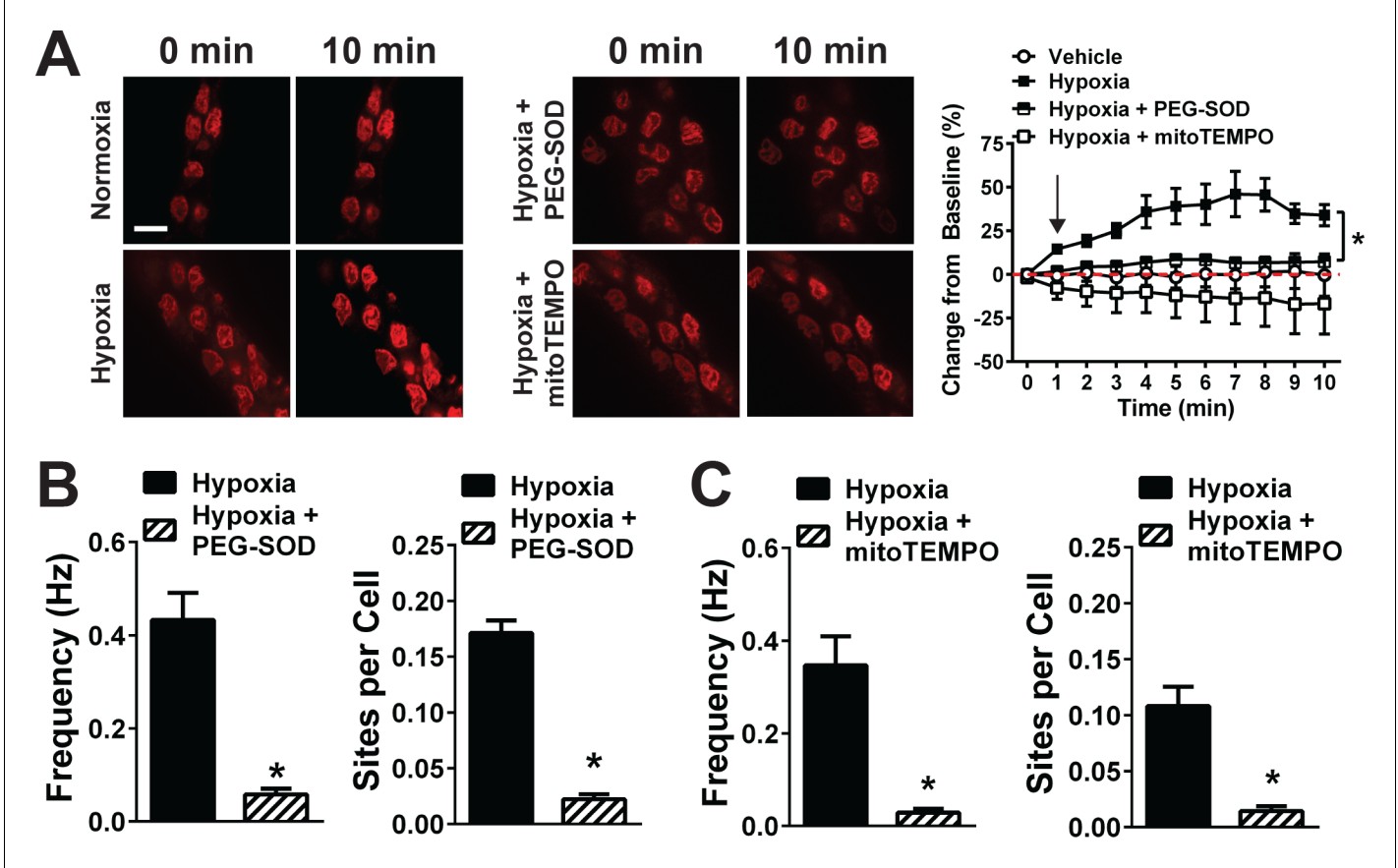

**Figure 4.** Acute hypoxia induces mitochondrial superoxide generation and increases TRPA1 sparklet frequency in the cerebral endothelium. (**A**) *Left and middle:* Superoxide generation detected with the superoxide-sensitive dye DHE (red) in the endothelium of cerebral arteries mounted *en face* under normoxia (left), hypoxia (left), hypoxia plus the membrane-permeable PEG-SOD (100 U/ml, middle) and hypoxia plus the mitochondria-targeted SOD mimetic mitoTEMPO (500 nM, middle). *Right:* Summary data showing changes in DHE fluorescence over time under each condition (*$p<0.05$, two-way ANOVA; n = 5–7 – 6–7 arteries from five different mice). The arrow in the graph indicates the onset of hypoxia. Scale bar = 10 μm. (**B and C**) Summary data showing the effects of PEG-SOD (**B**) and mitoTEMPO (**C**) on the frequency and number of active TRPA1 sparklet sites in the cerebral artery endothelium of *Tek:Gcamp6f* mice (*$p<0.05$, *Student*'s t-test; n = 25–30 fields of view from six different preparations isolated from six different mice). TRPA1 sparklets were recorded in the presence of EGTA-AM (10 μM) and (CPA, 30 μM).

DOI: https://doi.org/10.7554/eLife.35316.030

The following source data and figure supplements are available for figure 4:

**Source data 1.** Excel spreadsheet containing the individual numeric values of the parameters analyzed in *Figure 4*.
DOI: https://doi.org/10.7554/eLife.35316.033

**Figure supplement 1.** NOX2 inhibition and extracellular SOD did not significantly inhibit hypoxia-induced TRPA1 sparklets.
DOI: https://doi.org/10.7554/eLife.35316.031

**Figure supplement 1—source data 1.** Excel spreadsheet containing the individual numeric values of the parameters analyzed in *Figure 4—figure supplement 1*.
DOI: https://doi.org/10.7554/eLife.35316.032

MCAO, significantly reduced infarct size in control C57/bl6 mice (*Figure 8B*). This protective effect of CinA is partially dependent on endothelial cell TRPA1 channels as infarct size in *Trpa1* ecKO mice was not significantly reduced by CinA (*Figure 8C*). These data suggest that the endogenous activity of TRPA1 channels in the cerebral endothelium reduces cerebral damage associated with ischemic strokes.

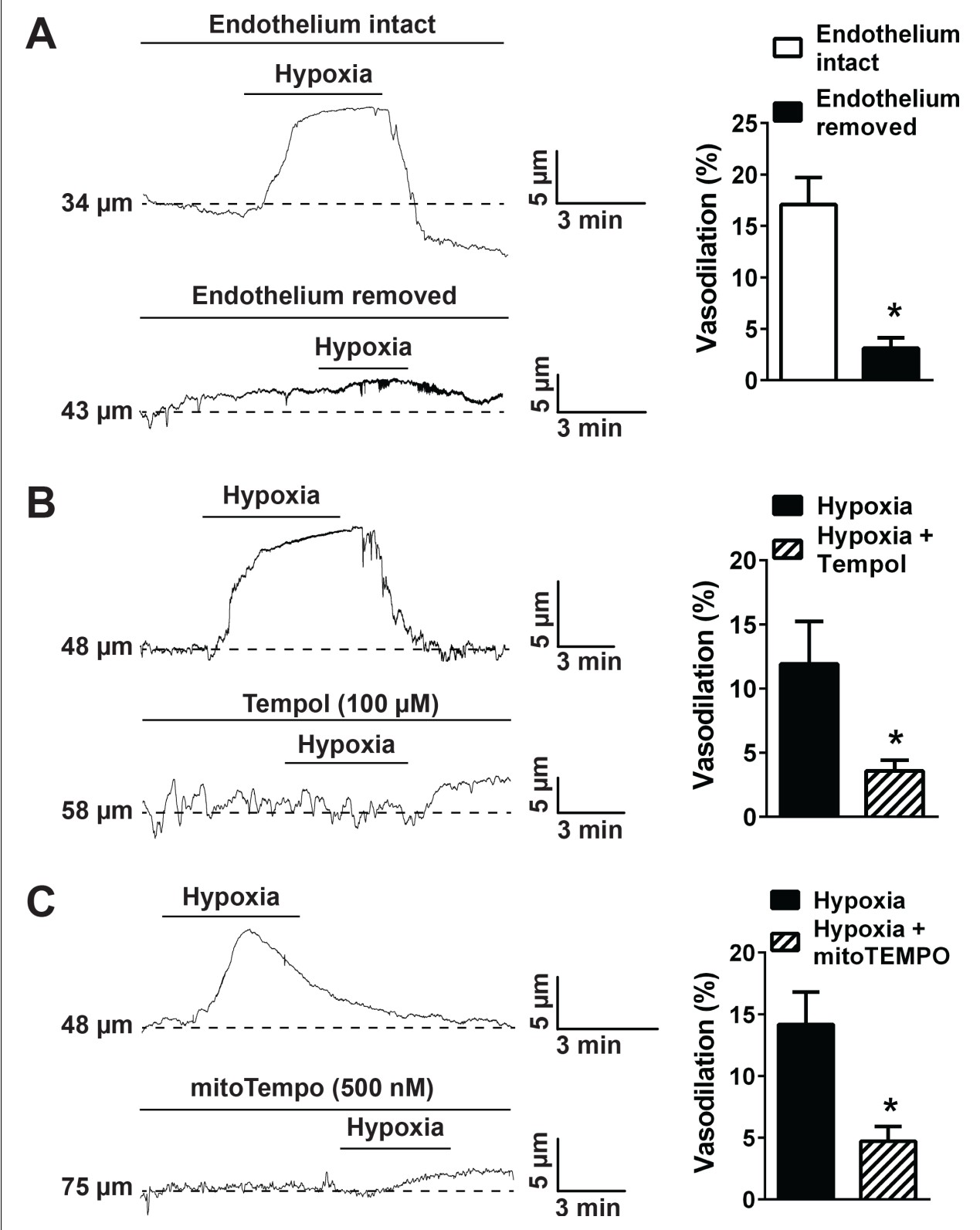

**Figure 5.** Hypoxia acts *via* mitochondrial superoxide to induce endothelium-dependent dilation of pressurized cerebral arteries. (A) Representative traces (left) and summary data (right) showing hypoxia-induced dilation in intact and endothelium-denuded cerebral pial arteries (*p<0.05, *Student*'s t-test; n = 5 arteries from three different mice). (B) Representative traces of lumen diameter of a pressurized cerebral pial artery (left) and summary data (right) showing hypoxia-induced dilation in the presence of the cell-permeant SOD mimetic Tempol (100 μM). (*p<0.05, *Student*'s t-test; n = 5 arteries

*Figure 5 continued on next page*

*Figure 5 continued*

from three different mice.) (**C**) Representative traces of the lumen diameter of a pressurized cerebral pial artery (left) and summary data (right) showing hypoxia-induced dilation in the presence of the cell-permeant mitochondrial membrane-targeted SOD mimetic mitoTEMPO (500 nM). (*p<0.05, *Student*'s t-test; n = 6 arteries from three different mice). The arteries used for pressure myography experiments were not treated with EGTA-AM or CPA.

DOI: https://doi.org/10.7554/eLife.35316.034

The following source data and figure supplements are available for figure 5:

**Source data 1.** Excel spreadsheet containing the individual numeric values of the parameters analyzed in *Figure 5*.

DOI: https://doi.org/10.7554/eLife.35316.039

**Figure supplement 1.** NOX2 inhibition and quenching of extracellular $O_2^-$ and $H_2O_2$ did not significantly alter cerebral artery dilation induced by hypoxia.

DOI: https://doi.org/10.7554/eLife.35316.035

**Figure supplement 1—source data 1.** Excel spreadsheet containing the individual numeric values of the parameters analyzed in *Figure 5—figure supplement 1*.

DOI: https://doi.org/10.7554/eLife.35316.036

**Figure supplement 2.** Superoxide dismutase mimetics do not directly inhibit TRPA1 channels.

DOI: https://doi.org/10.7554/eLife.35316.037

**Figure supplement 2—source data 1.** Excel spreadsheet containing the individual numeric values of the parameters analyzed in *Figure 5—figure supplement 2*.

DOI: https://doi.org/10.7554/eLife.35316.038

## Discussion

In this study, we investigated the functional importance of TRPA1 channels in the cerebral endothelium under pathophysiological conditions. We found that acute hypoxic exposure induced an increase in TRPA1 sparklet frequency in the endothelium of intact cerebral pial arteries and penetrating arterioles that caused dilation. Pharmacological activation of TRPA1 in vivo reduced the loss of brain tissue in response to experimentally induced ischemic stroke, whereas conditional knockout of TRPA1 in the endothelium exacerbated this damage (*Figure 8—figure supplement 3*). We propose that TRPA1 activity is important in mediating hypoxia-induced dilation of the cerebral vasculature during ischemic stroke, and that this response improves collateral blood flow to the affected region to improve outcomes.

TRPA1 channels are activated by electrophilic compounds, such as AITC, allicin and CinA, which target specific cysteine residues located in the cytosolic pre-S1 domain linking the channel's ankyrin-repeat region to the first membrane-spanning domain (*Hinman et al., 2006*; *Macpherson et al., 2007*; *Paulsen et al., 2015*). In addition, considerable evidence demonstrates that TRPA1 channels are activated by ROS and/or ROS-derived metabolites (*Sullivan et al., 2015*; *Trevisani et al., 2007*; *Pires and Earley, 2017*). Lipid peroxidation products such as 4-HNE and related substances appear to act on the same cysteine residues that are targeted by electrophilic molecules (*Hinman et al., 2006*; *Macpherson et al., 2007*; *Pires and Earley, 2017*), providing evidence for a common mechanism of activation. It has been suggested that TRPA1 channels are activated by hypoxia, such as reported by Takahashi *et al.*, who showed that shifting $pO_2$ from 150 mmHg to 80 mmHg stimulated TRPA1 activity, measured using the FLIPPER assay in HEK cells overexpressing TRPA1. The authors interpreted this finding as evidence of hypoxia-induced activation of TRPA1. However, under normal physiological conditions in healthy humans, the $pO_2$ of arterial blood is ~80–100 mmHg within the aorta,~30–70 mmHg in capillaries, and ~20–40 mmHg in the venous system (*Tsai et al., 2003*). Viewed in this context, the data reported by Takahashi *et al.* appear to suggest that TRPA1 channels are activated by restoration of normoxia following hyperoxic exposure, rather than by hypoxia per se. In our study, the $pO_2$ of the superfusing bath was closely monitored and was maintained at ~80–90 mmHg during control (normoxic) conditions and determined to be ~13 mmHg during acute hypoxic challenge. TRPA1 activity, detected as TRPA1 sparklets, was low under normoxic conditions, but was stimulated by change to a hypoxic $pO_2$. It should be noted that the $pO_2$ in the rat brain following an ischemic stroke varies from near anoxia in the infarct core (~1–2 mmHg) to ~10–15 mmHg in the peri-infarct area (*Liu et al., 2004*) — values of $pO_2$ similar to those observed in our tissue bath experiments. Thus, our data suggest that pathophysiologically relevant levels of hypoxia acutely activate TRPA1 channels in the intact cerebral endothelium.

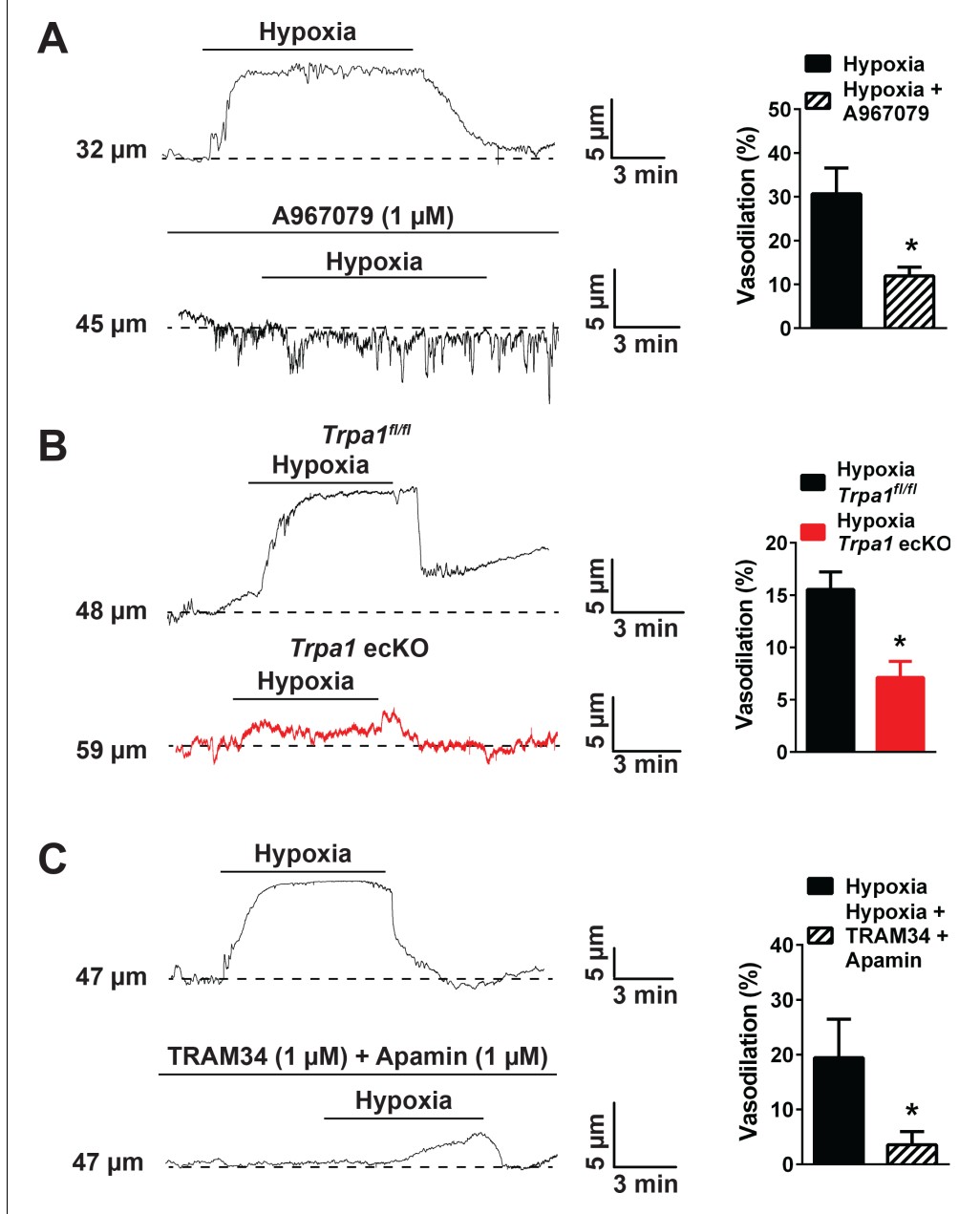

**Figure 6.** Hypoxia-induced cerebral artery dilation is dependent on endothelial TRPA1, $K_{Ca}3.1$ and $K_{Ca}2.3$ channels. (**A**) Representative traces of luminal diameter of pressurized cerebral pial arteries (left) and summary data (right) showing hypoxia-induced dilation in the presence and absence of the selective TRPA1 inhibitor A967079 (*$p < 0.05$, *Student*'s t-test; n = 5 arteries from three different mice). (**B**) Representative traces of the luminal diameter of pressurized cerebral pial arteries (left) and summary data (right) showing hypoxia-induced dilation in *Trpa1* ecKO mice and wildtype littermates (*Trpa1*$^{fl/fl}$) (*$p < 0.05$, *Student*'s t-test; n = 7–6 arteries from three different mice). (**C**) Representative traces of the luminal diameter of pressurized cerebral pial arteries (left) and summary data (right) showing hypoxia-induced dilation in the presence of selective inhibitors of $K_{Ca}3.1$ (TRAM34, 1 µM) and $K_{Ca}2.3$ channels (apamin, 1 µM) (*$p < 0.05$, *Student*'s t-test; n = 5 arteries from three different mice). The arteries used for pressure myography experiments were not treated with EGTA-AM or CPA.

DOI: https://doi.org/10.7554/eLife.35316.040

The following source data is available for figure 6:

**Source data 1.** Excel spreadsheet containing the individual numeric values of the parameters analyzed in *Figure 6*.
DOI: https://doi.org/10.7554/eLife.35316.041

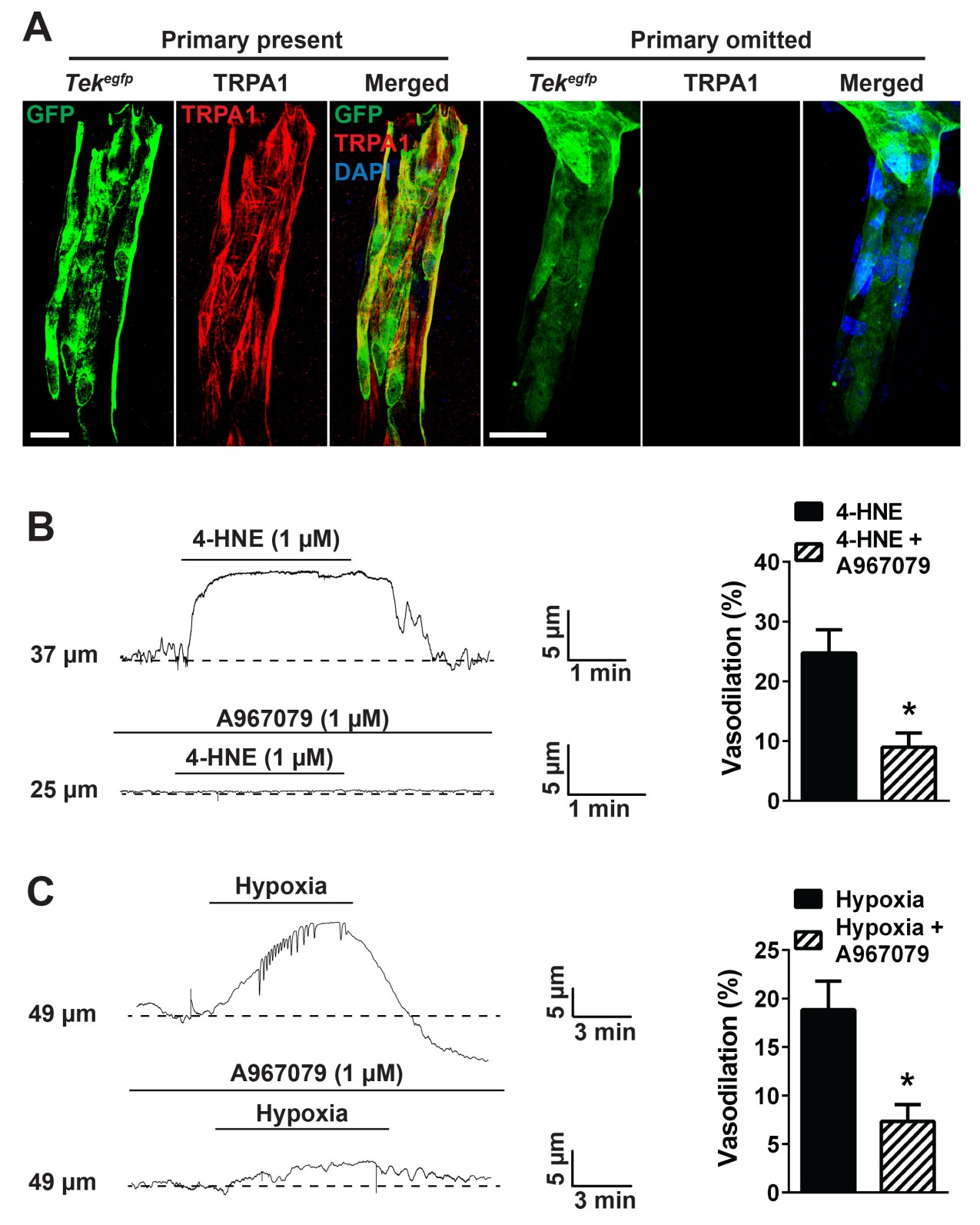

**Figure 7.** TRPA1 channel activity dilates cerebral penetrating arterioles. (A) Representative maximum intensity projection laser scanning confocal images showing immunolabeling of TRPA1 channels (red) in EGFP-expressing endothelial cells (green) of penetrating arterioles in the brain (left panels, primary present). Scale bar = 20 μm. TRPA1 immunoreactivity was absent when the primary antibody was omitted (right panels). Nuclei of cells were stained by DAPI (blue). Scale bar = 20 μm. (B) Incubation of ex vivo pressurized penetrating arterioles with the TRPA1 agonist 4-HNE (1 μM) induced

*Figure 7 continued on next page*

*Figure 7 continued*

arteriolar dilation which was significantly diminished by A967079 (1 μM). Representative traces are shown on the left and the summary graph is shown on the right. (*p<0.05, *Student*'s t-test; n = 5 arteries from three different mice). (C) Hypoxia induced dilation of penetrating arterioles that was significantly blunted by A967079 (*p<0.05, *Student*'s t-test; n = 6 arteries from three different mice). The arterioles used for pressure myography experiments were not incubated with EGTA-AM or CPA.

DOI: https://doi.org/10.7554/eLife.35316.042
The following source data is available for figure 7:

**Source data 1.** Excel spreadsheet containing the individual numeric values of the parameters analyzed in *Figure 7*.
DOI: https://doi.org/10.7554/eLife.35316.043

In a prior study, we showed that extracellular $O_2^-$ generated by NOX2 activates TRPA1 channels in the cerebral endothelium through a process that requires lipid peroxidation (*Sullivan et al., 2015*). However, here we found that hypoxia-induced activation of TRPA1 was independent of both NOX activity and extracellular $O_2^-$. Instead, we found that intracellular generation of $O_2^-$ was required for hypoxia-induced activation of TRPA1. Previous reports have shown that acute hypoxic exposure uncouples the mitochondrial electron transport chain (*Klimova and Chandel, 2008*), stimulating a localized burst of $O_2^-$ (*Hernansanz-Agustín et al., 2014*), which can enter the cytoplasm through anion channels present in the mitochondrial outer membrane (*Han et al., 2003*). In agreement with earlier reports showing that hypoxia increases mitochondrial $O_2^-$ production in cancer cells (*Chandel et al., 2000*; *Sabharwal and Schumacker, 2014*; *Eales et al., 2016*) and cultured endothelial cells (*Hernansanz-Agustín et al., 2014*), we found that exposure to acute hypoxia stimulated ROS generation in native cerebral artery endothelial cells, and that this response was inhibited by a mitochondrial-targeted SOD mimetic and by intracellular PEG-SOD. A recent study reported that mitochondria are located at the base of myoendothelial projections (*Maarouf et al., 2017*), which are regions of near contact between the plasma membrane of endothelial cells and the underlying smooth muscle cells. TRPA1 channels are also concentrated within myoendothelial projection (*Sullivan et al., 2015*; *Earley et al., 2009*). Thus, it is possible that hypoxia-induced increases in mitochondria $O_2^-$ production generates 4-HNE within myoendothelial projections, leading to TRPA1 activation in those sites. Our data provide support for a signaling pathway initiated by mitochondrial-generated $O_2^-$, showing that these superoxide ions stimulate the activity of TRPA1 channels in the cerebral endothelium during acute hypoxic exposure. The current studies do not specifically establish whether mitochondrial $O_2^-$ activates TRPA1 channels directly or if activation requires lipid peroxidation.

TRPA1 channels conduct mixed cation currents with a large $Ca^{2+}$ fraction (*Karashima et al., 2010*; *Wang et al., 2008*), and activation of TRPA1 on the plasma membrane of endothelial cells transiently creates small microdomains with high localized $Ca^{2+}$ concentrations (*Sullivan et al., 2015*). Our prior studies demonstrated that TRPA1-mediated $Ca^{2+}$ influx stimulated by direct application of AITC (*Earley et al., 2009*) or through $O_2^-$ generated by NOX2 (1) triggers endothelium-dependent dilation of cerebral arteries, a response that is unaffected by blocking nitric oxide synthase or cyclooxygenase pathways, but is sensitive to inhibition of $Ca^{2+}$-activated $K_{Ca}3.1$ and $K_{Ca}2.3$ $K^+$ channels. TRPA1 channels and $K_{Ca}3.1$ channels co-localize within myoendothelial projections (*Earley et al., 2009*), such that influx of $Ca^{2+}$ though TRPA1 channels within these spatially restricted regions is sufficient to increase the local $[Ca^{2+}]$ to levels capable of activating outward $K^+$ currents through $K_{Ca}3.1$ and $K_{Ca}2.3$, leading to hyperpolarization of the endothelial cell plasma membrane (*Sullivan et al., 2015*; *Earley et al., 2009*). Hyperpolarization of the endothelial cell plasma membrane, in turn, is conducted to SMCs through myoendothelial gap junctions, resulting in relaxation and vasodilation (*Sokoya et al., 2006*; *Chadha et al., 2011*). In addition, TRPA1 activation by AITC (*Qian et al., 2013*) and 4-HNE stimulates $Ca^{2+}$ release from intracellular stores, leading to an increase in $Ca^{2+}$ transients previously linked to activation of $K_{Ca}3.1$ (39). Our current findings show that activity of $K_{Ca}3.1$ and/or $K_{Ca}2.3$ channels is required for cerebral artery dilation in response to hypoxia, suggesting that these channels act downstream of TRPA1 in this setting.

Our data also show that pharmacological inhibition of TRPA1 channels or *Trpa1* knockout in the endothelium did not completely abolish hypoxia-induced dilation of cerebral arteries and arterioles, suggesting the possibility that redundant or overlapping mechanisms contribute to this response.

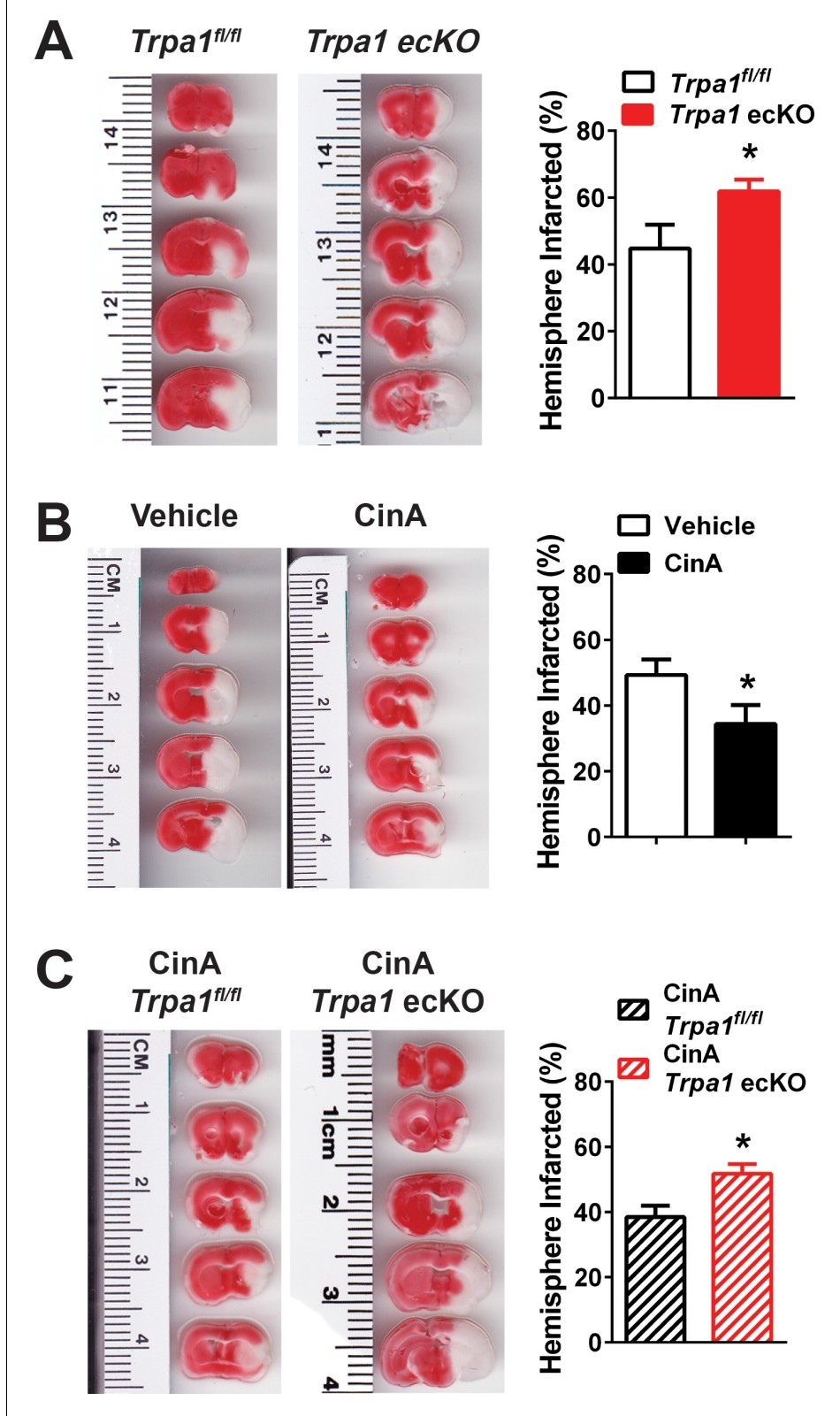

**Figure 8.** Endothelial cell TRPA1 channel activity protects against ischemic strokes. (**A**) Representative photographs of brain slices (left) and summary data (right) showing significantly greater ischemic damage 24 hr after MCAO in *Trpa1* ecKO mice compared with *Trpa1^fl/fl*. Brain slices were stained with 2,3,5-triphenyltetrazolium chloride (TTC), which stains metabolically active tissue red, whereas infarcted tissue remains unstained (white). Infarcted areas were quantified and expressed as a percentage of total hemisphere area (*p<0.05, *Student*'s t-test; n = 5–5 mice). (**B**) Representative
*Figure 8 continued on next page*

*Figure 8 continued*

photographs of brain slices (left) and summary data (right) showing reduced cerebral ischemic damage in wildtype C57/bl6 mice treated with the TRPA1 channel activator cinnamaldehyde (CinA, 50 mg/kg i.p.), injected 15 min after MCAO (*p<0.05 for CinA vs. vehicle, *Student*'s t-test; n = 5–6 mice). (C) Representative photographs of brain slices (left) and summary data (right) showing that the protective effects of CinA were blunted in *Trpa1* ecKO mice (*p<0.05 for CinA-treated *Trpa1* ecKO mice vs. CinA-treated *Trpa1*$^{fl/fl}$ mice, *Student*'s t-test; n = 6–5 mice). Legends for Supplemental Figures.

DOI: https://doi.org/10.7554/eLife.35316.044

The following source data and figure supplements are available for figure 8:

**Source data 1.** Excel spreadsheet containing the individual numeric values of the parameters analyzed in *Figure 8*.
DOI: https://doi.org/10.7554/eLife.35316.050

**Figure supplement 1.** Post-MCAO perfusion is not significantly different between all experimental groups.
DOI: https://doi.org/10.7554/eLife.35316.045

**Figure supplement 1—source data 1.** Excel spreadsheet containing the individual numeric values of the parameters analyzed in *Figure 8—figure supplement 1*.
DOI: https://doi.org/10.7554/eLife.35316.046

**Figure supplement 2.** Cinnamaldehyde causes cerebral artery and arteriolar dilation by activating TRPA1 channels in endothelial cells.
DOI: https://doi.org/10.7554/eLife.35316.047

**Figure supplement 2—source data 1.** Excel spreadsheet containing the individual numeric values of the parameters analyzed in *Figure 8—figure supplement 2*.
DOI: https://doi.org/10.7554/eLife.35316.048

**Figure supplement 3.** Mechanism of endothelial TRPA1 channel-mediated neuroprotection.
DOI: https://doi.org/10.7554/eLife.35316.049

Several possibilities have been reported, including endogenous generation of carbon monoxide (*Leffler et al., 1999*) and TRPV3 channel activity. TRPV3 has been shown to be sensitized by acute hypoxia in a ROS-independent manner (*Karttunen et al., 2015*), and cause endothelium-dependent dilation of cerebral pial arteries and penetrating arterioles in response to chemical agonists (*Pires et al., 2015*; *Earley et al., 2010*).

Ischemic stroke is one of the leading causes of death and disability worldwide. The extent of neuronal loss following an ischemic insult is determined in part by the supply of blood to the region surrounding the infarct core, known as the ischemic penumbra (*Maas et al., 2009*; *Hoehn-Berlage et al., 1995*). The penumbra is characterized by structurally intact, but metabolic silent, parenchymal tissue in which perfusion is impaired (*Symon et al., 1977*). Increasing blood flow to the ischemic penumbra has been shown to reduce expansion of the infarct zone and, consequently, improve neurological outcomes (*Cipolla et al., 2018*). Our findings suggest that activation of TRPA1 channels in endothelial cells of cerebral arteries is an early adaptive response to acute reduction in tissue $pO_2$, leading to a possible increase in perfusion within the penumbra. Loss of TRPA1 signaling in endothelial cells of cerebral arteries increased cerebral infarcts following permanent middle cerebral artery occlusion in mice. Further, pharmacological activation of TRPA1 decreased infarcts, an effect blunted by loss of endothelial TRPA1 channels. Isoflurane, the anesthetic agent used during MCAO surgeries, was previously shown to potentiate TRPA1 activity (*Matta et al., 2008*). Interestingly, isoflurane anesthesia is neuroprotective in various animal models of stroke and subarachnoid hemorrhage (*Altay et al., 2012a*; *Altay et al., 2012b*; *Li et al., 2013*; *Khatibi et al., 2011*), but the mechanistic basis of this effect is not known. The current findings support the concept that isoflurane may provide neuroprotection by potentiating TRPA1-mediated cerebral arterial dilation.

In summary, the present study shows that hypoxia stimulates $Ca^{2+}$ influx through TRPA1 channels in the intact cerebral endothelium *via* a mechanism that requires generation of intracellular $O_2^-$ by mitochondria. We also demonstrated that hypoxia-induced TRPA1 activity initiates endothelium-dependent dilation of cerebral pial arteries and penetrating arterioles. Selective deletion of TRPA1 expression in the endothelium exacerbated the loss of brain tissue associated with ischemic stroke, providing evidence that hypoxia-induced activation of TRPA1 in the cerebral endothelium constitutes a novel adaptive response.

# Materials and methods

## Key resources table

| Reagent type (species) or resource | Designation | Source or reference | Identifiers | Additional information |
|---|---|---|---|---|
| strain, strain background (*Mus musculus*), C57bl6 | *Tek^cre* | Jackson Laboratories; PMID: 11161575 | stock # 008863 | |
| strain, strain background (*Mus musculus*), C57bl6 | *mT/mG* | Jackson Laboratories; PMID: 17868096 | stock # 007676 | |
| strain, strain background (*Mus musculus*), C57bl6 | *Tek^gfp* | Jackson Laboratories; PMID: 11064424 | stock # 003658 | |
| strain, strain background (*Mus musculus*), C57bl6 | *Gcamp6f* | Jackson Laboratories; PMID: 25741722 | stock # 028865 | |
| strain, strain background (*Mus musculus*), C57bl6 | *Trpa1 ecKO* | Other; PMID: 25564678 | NA | Mice with endothelium-specific deletion of Trpa1. We have characterized these mice in a previous publication (PMID: 25564678). |
| strain, strain background (*Mus musculus*), C57bl6 | *Tek:Gcamp6f* | This paper | NA | Mice expressing the fast kinetics, genetically encoded $Ca^{2+}$ biosensor *Gcamp6f* exclusively in endothelial cells. |
| antibody | anti-4-HNE (rabbit polyclonal) | Abcam | ab46545 | 1:1000 dilution |
| antibody | anti-TRPA1 (rabbit polyclonal) | Alomone Labs; UniProtKB - O75762 (TRPA1_HUMAN) | ACC-037 | 1:1000 dilution |
| antibody | anti-GFP (goat polyclonal) | Abcam; UniProtKB - P42212 (GFP_AEQVI) | ab5450 | 1:500 dilution |
| antibody | Alexa 488 or 594 secondaries | ThermoFisher Scientific | A11055 (Alexa 488); A21207 (Alexa 594) | 1:1000 (Alexa 488); 1:2000 (Alexa 594) |
| other | Isolectin GS-IB$_4$ conjugated to Alexa 488 | ThermoFisher Scientific | I21411 | 1:1000 dilution |
| other | dyhydroethidium | ThermoFisher Scientific | D11347 | |
| other | Fluoroshied mounting medium with DAPI | Abcam | ab104139 | |
| software, algorithm | SparkAn | Dr. Adrian Bonev and Dr. Mark Nelson; PMID: 22095728 | NA | Software to analyze $Ca^{2+}$ events. Kindly provided by Dr. Adrian Bonev and Dr. Mark Nelson from the University of Vermont. |

## Animals

Adult male and female mice (12–16 weeks of age) were used for all experiments. All animal procedures used in this study were approved by the Institutional Animal Care and Use Committee of the University of Nevada, Reno School of Medicine, and are in accordance with the National Institutes of Health 'Guide for the Care and Use of Laboratory Animals', eigth edition.

## Tek:Gcamp6f mice

Endothelial cell-specific expression of the fast kinetics, genetically encoded $Ca^{2+}$ biosensor *Gcamp6f* (*Chen et al., 2013*) in mice was achieved by crossing mice heterozygous for expression of a floxed *Stop* codon upstream of the *Gcamp6f* gene (Jackson Labs, stock number: 028865) with mice

heterozygous for the expression of *cre* recombinase under the control of the endothelial-specific *Tek* promoter/enhancer (*Tek^cre*; Jackson Labs, stock number: 008863). We designated this strain *Tek: Gcamp6f*. We confirmed that *cre*-recombinase is only expressed in the endothelium of cerebral arteries of *Tek^cre* mice by crossing them with the reporter strain *mT/mG* mice (Jackson Labs, stock number: 007676). *mT/mG* mice have loxP sites flanking the genetic sequence for the red fluorescent protein *Tomato* and a *Stop* codon; the genetic sequence for the green fluorescent protein *Egfp* is located downstream from the second loxP site (see diagram in *Figure 1—figure supplement 1*). Thus, expression of *cre*-recombinase removes the sequence for *Tomato* and the *Stop* codon, leading to Egfp production in that particular cell type (*Muzumdar et al., 2007*). Using this reporter strain, we observed that only endothelial cells expressed Egfp, whereas the underlying smooth muscle layer expressed the red fluorescent protein Tomato (*Figure 1—figure supplement 1*).

### Trpa1 ecKO mice

Endothelial cell-specific deletion of Trpa1 was achieved by crossing mice homozygous for floxed Trpa1 channels containing loxP sites flanking S5/S6 transmembrane domains (Jackson Labs, stock number: 008654) with heterozygous *Tek^cre* mice, to generate *Trpa1* ecKO mice, as we have previously described (*Sullivan et al., 2015*). Mice homozygous for floxed Trpa1, but without expression of cre-recombinase (*Trpa1^{fl/fl}*) were used as controls for all experiments.

## Recording and analysis of TRPA1 sparklets

For isolation of cerebral arteries, mice were deeply anesthetized with 4% isoflurane and euthanized by decapitation and exsanguination. The skull was carefully removed and the brain with the brainstem intact was placed in ice-cold tissue collection solution consisting of (in mM) 140 NaCl, 5 KCl, 2 $MgCl_2$, 10 dextrose, 10 HEPES (pH 7.4), without $Ca^{2+}$, supplemented with 1% bovine serum albumin (BSA). Cerebral pial arteries were carefully removed from the brain and cleaned of meningeal membranes. $Ca^{2+}$ imaging was performed on cerebral arteries isolated from *Tek:Gcamp6f* mice mounted *en face*. Arteries were opened longitudinally using fine spring scissors (Fine Science Tools, Foster, CA) and mounted on a Sylgard pad using insect pins with the endothelium facing up as previously described (*Sonkusare et al., 2012*; *Ledoux et al., 2008*). The tissue was stretched to its in vivo length and maintained in $Ca^{2+}$-free PSS containing the membrane-permeable $Ca^{2+}$ chelator EGTA-AM (10 µM, ThermoFisher Scientific, Eugene, OR) at 37°C for 15 min, then washed with warm (37°C) $Ca^{2+}$ PSS consisting of (in mM): 119 NaCl, 4.7 KCl, 21 $NaHCO_3$, 1.18 $KH_2PO_4$, 1.17 $MgSO_4$, four dextrose, and 1.8 $CaCl_2$. The solution was continuously aerated with a normoxic gas mixture (21% $O_2$, 6% $CO_2$, 73% $N_2$) to maintain a constant pH of 7.38–7.42 and $pO_2$ of ~80 mmHg. The pH was continuously measured with an in-line flow-through pH mini-electrode, and solution oxygenation was assessed using an in-line oxygen microelectrode (both from Microelectrodes, Inc., Bedford, NH). ER $Ca^{2+}$ stores were depleted by treating the preparation with the SERCA inhibitor CPA (30 µM; Tocris Biosciences, Bristol, UK) to eliminate $Ca^{2+}$ signals generated by spontaneous ER $Ca^{2+}$ release. CPA was maintained in the bath solution throughout the experiment. All pharmacological agents (4-HNE, A967079, PEG-SOD, mitoTEMPO, extracellular SOD and apocynin) were administered via the superfusing bath solution. Hypoxia was induced by superfusing the tissue with PSS aerated with a hypoxic gas mixture (5% $O_2$, 6% $CO_2$, 89% $N_2$). The $pO_2$ of this solution was 13 ± 2 mmHg, and the pH in the recording chamber was 7.3–7.42. $Ca^{2+}$ images were obtained using an inverted microscope (Olympus iX81; Olympus Corp., Tokyo, Japan), modified to allow imaging in the upright configuration (LSM Tech, Etters, PA) and equipped with epifluorescence illumination, a 60x water-immersion objective (numerical aperture 1.0), and a highly sensitive iXon Ultra EMCCD camera (Andor Technology, Belfast, Northern Ireland). Each field of view was 512 × 512 pixels (one pixel = 0.27 µm), and images were recorded at a rate of 30–55 frames/s for a total of 1000 frames. The duration of each recording was 20–30 s. A separate cohort of *Tek:Gcamp6f mice* were used to assess the effects of 4-HNE on $Ca^{2+}$ signaling activity in the cerebral endothelium in the absence of EGTA-AM and CPA.

Recordings were analyzed using custom software (SparkAn, kindly provided by Drs. Adrian Bonev and Mark T. Nelson, University of Vermont, Burlington Vermont) (*Dabertrand et al., 2012*). An average of the first 10 images of each recording obtained prior to stimulation was used to define baseline fluorescence ($F_0$). TRPA1 sparklets were identified as localized increases in fluorescence ($\Delta F$) within a small region of interest (5.3 × 5.3 µm) after subtraction of $F_0$, and manifested as quantal

events with characteristic step-wise peaks in plots of fluorescence intensity over time. The frequency of $Ca^{2+}$ signals with amplitude between 1.08 and 1.1 $\Delta F/F_0$ (and multiples) is reported. Each peak was analyzed individually, and amplitude, attack time (half-time elapsed from the beginning of the event to the peak), decay time (half-time elapsed from the peak to dissipation of the event), duration (total open time of the channel, calculated as [attack + decay] x 2), and spatial spread of the $Ca^{2+}$ signal were determined.

## Immunofluorescence labeling

Accumulation of 4-HNE in cerebral arteries exposed to hypoxia was assessed by immunofluorescence labeling. Freshly isolated cerebral arteries were mounted *en face* as described above, and superfused with normoxic or hypoxic PSS for 15 min. The preparations were then fixed with 4% formaldehyde in PBS for 10 min and incubated in 5% BSA +0.1% Triton X-100 for 2 hr at room temperature. Preparations were then incubated with a rabbit polyclonal antibody against 4-HNE (1:1000, Abcam PLC, Cambridge, United Kingdom) overnight. Preparations were washed with PBS and incubated with a donkey anti-rabbit secondary antibody conjugated to AlexaFluor 568 (1:2000, Thermo Fisher Scientific) diluted in 2% BSA +0.1% Triton X-100 for 90 min at room temperature. Preparations were then washed with PBS and incubated with the endothelial cell-labeling isolectin GS-IB4 conjugated to AlexaFluor 488 (1:1000 in PBS, Thermo Fisher Scientific) for 10 min at room temperature. Preparations were washed and mounted in glass slides with coverslips and fluorescence analyzed by laser scanning confocal microscopy in an Olympus FluoView confocal microscope within the same day. Images were obtained in a field of view of 800 × 800 pixels using a 60x oil immersion objective (numerical aperture: 1.42) and a Z-stack spanning the entire thickness of the cerebral artery, from the adventitia to the endothelial cell layer. In order to perform the semi-quantitative assessment of 4-HNE fluorescence, settings (PMT sensitivity, gain and laser power) were adjusted for normoxic cerebral arteries, and the same settings were used for cerebral arteries exposed to hypoxia. Quantification of fluorescence intensity was performed by plotting a Z-axis profile on ImageJ to obtain a fluorescence intensity curve, and the area under the curve was calculated to best represent fluorescence intensity of the entire thickness of the preparation.

A separate group of mice were used to determine if TRPA1 channels are present in penetrating arterioles in the brain. Endothelial cell reporter mice constitutively expressing GFP under the control of the Tek promoter ($Tek^{gfp}$, Jackson labs, stock number 003658) were perfusion-fixed with 4% formaldehyde in PBS and the brain was removed from the skull for histological processing. Brain slices (200 µm thick) were acquired using a Leica VT 1200S vibratome (Leica Biosystems GmbH, Wetzlar, Germany) and permeabilized with 0.1% Triton-X in PBS for 2 hr at room temperature. Slices were then incubated in 10% normal horse serum +0.1% Triton-X in PBS for 2 hr at room temperature, followed by incubation with the primary antibodies rabbit anti-TRPA1 (1:1000, Alomone Labs, Jerusalem, Israel) and goat anti-GFP (1:500, Abcam) dissolved in blocking solution overnight at room temperature on a shaker. Controls lacking TRPA1 primary antibody were incubated with anti-GFP alone overnight. Slices were washed with PBS 3 times for 10 min each, then incubated with a donkey anti-rabbit conjugated with AlexaFluor 594 (1:2000, Thermo Fisher Scientific) and a donkey anti-goat conjugated with AlexaFluor 488 (1:1000, Thermo Fisher Scientific) for 2 hr at room temperature on a shaker. Slices were washed three times with PBS and mounted on glass slides using a Fluoroshield Mounting Medium with DAPI (Abcam). Fluorescence images were taken by an Olympus FluoView laser scanning confocal microscope using a 60x oil-immersion objective (numerical aperture: 1.42) in a 1024 × 1024 pixels per field of view. The pixel size was maintained constant at X = 180 nM, Y = 180 nM, Z = 0.801 nM. Z-stacks were captured (~40 µm in thickness) to allow for 3-dimensional reconstruction of the arteriolar bed.

## ROS imaging

Superoxide production by native cerebral artery endothelial cells was assessed by confocal imaging of cells stained with the ROS-sensitive dye dyhydroethidium (DHE) (*Pires et al., 2010*). Native endothelial cells were isolated by dissecting cerebral arteries and incubating them in tissue collection solution supplemented with 1 mg/mL papain (Worthington Biochemical Company, Lakewood, NJ), 1 mg/mL DL-dithiothreitol and 10 mg/mL BSA for 12 min in a 37°C water bath. Arteries were then washed three times with $Ca^{2+}$-free PSS and incubated in tissue-collection solution supplemented

with 1 mg/mL type II collagenase (Worthington Biochemical Company) for 12 min in a 37°C water bath. Thereafter, arteries were washed three times with PSS and triturated by passing the solution through a fire-polished glass Pasteur pipette, as described previously (*Pires et al., 2015*; *Pires et al., 2017*). Dissociated cells in the solution were placed in a 35 mm glass-bottom dish (Greiner Bio-One GmBH, Germany) and allowed to adhere to the dish for 30 min in a cell culture incubator. After incubating cells with 5 µM DHE (ThermoFisher) for 5 min at 37°C, the culture dish was transferred to a spinning-disk confocal microscope for real-time recordings of fluorescence intensity. Attached cells in the dish were exposed to normoxic or hypoxic PSS supplemented with DHE (5 µM) in cells pre-incubated with vehicle, PEG-SOD (30 min prior to recording) or mitoTEMPO (15 min prior to recording) and recorded at a rate of 1 frame/min for 10 min. Recordings were performed in an Olympus IX-71 microscope coupled to a Yokogawa CSU22 Confocal Scanning Unit (Yokogawa Electric Corporation, Tokyo, Japan) and an Andor iXon + camera (Andor Technology) using a 100x oil-immersion objective (numerical aperture 1.45). Cells were excited using a 488 nm laser and emission was detected using a 500–530 band-pass emission filter.

## Pressure myography

The effect of hypoxia on the diameters of isolated intact cerebral pial resistance arteries was assessed using pressure myography. Cerebral arteries were carefully dissected and cannulated between two glass cannulas in a pressure myograph chamber (Living Systems Instrumentation, St. Albans, VT). Arteries were pressurized to 60 mmHg and maintained in normoxic PSS at 37°C for 30 min to reach equilibration. Luminal diameter was continuously recorded by edge-detection using videomicroscopy. The viability of each preparation was established by briefly incubating arteries in isotonic PSS containing high extracellular $K^+$ (60 mM KCl, 59 mM NaCl; all other components held constant) to induce constriction. Afterwards, arteries were washed with normoxic PSS until spontaneous myogenic tone was generated. Myogenic tone (%) was calculated as [1 − (active luminal diameter/passive luminal diameter)] x 100. Arteries were then exposed to hypoxic PSS for 10 min and subsequently returned to normoxic PSS. All pharmacological experiments were performed in a paired fashion. Passive diameter at 60 mmHg was determined by incubating arteries in $Ca^{2+}$-free PSS supplemented with EGTA (2 mM) and diltiazem (10 µM). In a subset of experiments, the endothelium was removed by passing an air bubble through the lumen of the artery (*Ralevic et al., 1989*). This method of endothelium removal was shown to almost completely ablate endothelium-dependent responses, without damaging the underlying smooth muscle cell layer (*Ralevic et al., 1989*).

The effects of TRPA1 agonists and hypoxia were also assessed in freshly isolated cerebral penetrating arterioles. Isolation of penetrating arterioles was performed as previously described (*Pires et al., 2016*). Briefly, penetrating arterioles branching out of the middle cerebral artery were carefully isolated from the surrounding parenchyma and mounted onto glass cannulas in a blind-sac experimental configuration. Penetrating arterioles were pressurized to 40 mmHg and superfused with artificial cerebrospinal fluid (aCSF, in mM: 124 NaCl, 3 KCl, 2 $CaCl_2$, 2 $MgCl_2$, 1.25 $NaH_2PO_4$, 26 $NaHCO_3$, four glucose, pH 7.4). Arteriolar viability was assessed by exposing the preparation to isotonic aCSF containing high extracellular $K^+$ (in mM: 67 NaCl, 60 KCl, all other salts remain the same concentration). Viable arterioles were washed with regular aCSF and allowed to generate spontaneous myogenic tone, after which they were exposed to 4-HNE, hypoxia or CinA in the presence or absence of the TRPA1 inhibitor A967079.

## Middle cerebral artery occlusion model

Permanent cerebral ischemia was induced using the intraluminal suture model of MCAO (*Longa et al., 1989*), modified for mice. Mice were initially anesthetized with 2% isoflurane in oxygen and their body temperature was maintained at 37°C. An incision was made at the top of the head for attachment of a laser Doppler flow probe (Periflux System 5000; Perimed AB, Järfälla, Sweden) to measure blood flow to the region supplied by the MCA (5 mm lateral and 1 mm posterior to the bregma). A midline incision was made at the neck to expose the carotid artery, and the lingual, thyroid and external carotid arteries were tied off. A 6–0 nylon monofilament with a rounded silicone tip (Doccol Corporation, Sharon, MA),~210 µm in diameter, was inserted into the common carotid artery and advanced through the internal carotid artery to block the MCA where it branches from

the circle of Willis. MCAO was verified by a sharp drop in blood perfusion, as measured by laser Doppler flowmetry. Mice that showed less than an 80% drop in perfusion were excluded from the study. Ischemia was maintained for 24 hr, after which mice were anesthetized and euthanized by decapitation. A subset of mice was injected with CinA (50 mg/kg body weight [*Huang et al., 2007*]) or an equal volume of vehicle (0.2–0.3 ml of DMSO) 15 min after MCAO to assess the effects of pharmacological activation of TRPA1 on ischemic stroke outcome.

## Measurement of infarct size

The brain was removed from the skull and sliced into 1 mm-thick sections. Ischemic damage was assessed by subsequently staining sections for 20 min with 2% 2,3,5-triphenyltetrazolium chloride, which stains viable tissue red, leaving infarcted tissue white. Brain slices were fixed in 4% formaldehyde in PBS, and digital images were taken. The percentage of infarction was determined using the following equation: %HI = [(VCVL)/VC]*100, where HI is the hemisphere infarcted, VC is the volume of normal tissue in the non-ischemic hemisphere, and VL is the volume of normal tissue in the ischemic hemisphere (*Swanson et al., 1990*).

## Chemicals

Unless otherwise stated, all chemicals were purchased from Sigma-Aldrich Corporation (St. Louis, MO).

## Statistical analysis

All summary data are expressed as means ± SEM. Statistical analyses were performed and graphs were constructed using Prism 6.0 (GraphPad Software, Inc., La Jolla, CA). The significance of differences between two groups was tested by paired or unpaired two-tailed Student's t-test, depending on the experimental design. Data that did not fit a normal distribution were tested using a non-parametric alternative. Comparisons of three or more groups were tested by one-way analysis of variance (ANOVA) with a Sidak *post hoc* test. Values of $p < 0.05$ were considered statistically significant for all experiments.

Sample size for experiments were determined by performing a 2-sided power analysis to reach a power of 0.8 for a value of $\alpha \leq 0.05$. Using these parameters, we determined that $Ca^{2+}$ imaging experiments should have a minimum of 16 field of views; experiments assessing 4-HNE fluorescence should be 5 field of views; pressure myography experiments should be performed in at least three arteries; MCAO experiments required at least three mice.

## Acknowledgements

We thank Anita S. Savel for technical assistance and Drs. Mark Nelson and Adrian Bonev (University of Vermont) for kindly providing custom software (SparkAn) for the analysis of $Ca^{2+}$ signals.

## Additional information

### Funding

| Funder | Grant reference number | Author |
|---|---|---|
| American Heart Association | 15POST2472002 | Paulo Wagner Pires |
| National Heart, Lung, and Blood Institute | K99HL140106 | Paulo Wagner Pires Scott Earley |
| National Heart, Lung, and Blood Institute | R01HL137852 | Scott Earley |
| National Heart, Lung, and Blood Institute | R01HL091905 | Scott Earley |
| National Heart, Lung, and Blood Institute | R01HL139585 | Scott Earley |
| National Institute of Neurological Disorders and Stroke | RF1NS110044 | Scott Earley |

The funders had no role in study design, data collection and interpretation, or the decision to submit the work for publication.

## Author contributions
Paulo Wagner Pires, Conceptualization, Data curation, Formal analysis, Validation, Investigation, Visualization, Methodology, Writing—original draft, Project administration, Writing—review and editing; Scott Earley, Conceptualization, Resources, Data curation, Supervision, Funding acquisition, Validation, Visualization, Methodology, Project administration, Writing—review and editing

## Author ORCIDs
Paulo Wagner Pires http://orcid.org/0000-0001-5972-4554
Scott Earley http://orcid.org/0000-0001-9560-2941

## Ethics
Animal experimentation: All animal procedures used in this study were approved by the Institutional Animal Care and Use Committee of the University of Nevada, Reno School of Medicine (IACUC protocol #2016-00598), and are in accordance with the National Institutes of Health 'Guide for the Care and Use of Laboratory Animals', 8th edition.

## Decision letter and Author response
Decision letter https://doi.org/10.7554/eLife.35316.055
Author response https://doi.org/10.7554/eLife.35316.056

## Additional files

### Supplementary files
• Supplementary file 1. Table 1.
DOI: https://doi.org/10.7554/eLife.35316.051
• Supplementary file 2. Table 2.
DOI: https://doi.org/10.7554/eLife.35316.052
• Transparent reporting form
DOI: https://doi.org/10.7554/eLife.35316.053

### Data availability
All data generated or analyzed during this study are included in the manuscript and supporting files. Source data files have been provided for Figures 1 - 8 and Supplemental Figures 2,3,5,6,7,8,9 and 10.

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
