## [Decision Letter]

Thank you for submitting your article "Neuroprotective Effects of TRPA1 Channels in the Cerebral Endothelium Following Ischemic Stroke" for consideration by *eLife*. Your article has been reviewed by Richard Aldrich as the Senior Editor, a Reviewing Editor, and three reviewers. The reviewers have discussed the reviews with one another and the Reviewing Editor has drafted this decision to help you prepare a revised submission. The reviewers have opted to remain anonymous.

This is a potentially highly important paper that shows how oxidative species lead to vasodilation in pial brain vessels and potentially heightens blood flux during stress, injury, and/or hypoxia. While the reviewers found merit to your work, they also note serious complications. These are the essential issues that must be addressed in the text and/or with new data.

Past analysis by the Kleinfeld group and by various computational groups conclude that the bulk of the resistance of the vasculature lies in the capillaries with a smaller fraction in the penetrating arterioles. The current work considers only pial arterioles. At the minimum, it is essential to show that penetrating arterioles contain TRPA1 at densities similar or greater than in the pial arterioles. Ideally, the authors should isolate a pial vessel and show that these vessels also react to activation of TRPA1 by 4-HNE.

The reviewers raised many essential points. All the points listed below need to be addressed. The following are deemed essential and must be addressed with new text or additional experiments.

Essential revisions:

As noted by reviewer #1

"… it is therefore essential to carry out.… experiments showing that 4-HNE evokes detectable Ca^2+^ transients, that are mediated by TRPA1, in endothelial cells that have never been exposed to CPA or to EGTA-AM."

"There is an apparent contradiction in Figure 3A. The image shows increased DHE fluorescence with hypoxia, while the graph shows a decrease. Please clarify?

Further to Figure 3A, "It is essential to add data.… to demonstrate that there is a reduction of the generation of ROS inside the cells, as monitored with DHE, when PEG-SOD was applied."

As noted by reviewer #2

"[You] previously assigned the production of ROS to NADPH oxidase, yet the current work assigns it to mitochondria..… If TRPA1 is localized in endothelial projections near the smooth muscle contact site, how can ROS generated in mitochondria and outside projections, and diffuse to the channel without interacting with other proteins? Is it possible that TRPA1 lies outside the projection in the endothelial body?" Please clarify.

As noted by reviewer # 3

"Injection of CinA reduced the infarct size in control.… mice. It is not clear why they chose this to activate TRPA1 [as] there is no information that CinA acts TRPA1 to dilate cerebral arteries. Thus, it is important to see if CinA increases TRPA1 sparklets in wild type control [mice] but not in eTRPA1-/- [mice].

"[The] authors show that hypoxia produced 4-HNE.… However, it is not clear if 4-HNE contributes to TRPA1 sparklets. Based on the findings in Figure 3, TRPA1 channels were activated by mitochondrial ROS. Thus, it seems that 4-HNE do not have a major role in hypoxia-induced TRPA1 sparklets. Please clarify this issue in the Discussion.

Lastly, please make it clear in the Abstract that you are using isolated vessels for all but the infarct data. " Mouse model" is too nebulous a term.

Reviewer #1:

This paper reports that hypoxia of (pial) arteries leads to the generation of reactive oxygen species such as the lipid peroxide 4-hydroxynonenal (4-HNE), which can activate TRPA1 channels in the endothelial cell surface membrane. These let calcium in, which activates gK(Ca) channels, leading to a K^+^ efflux. This in turn raises [K]o around the arteries' smooth muscle, leading to hyperpolarization (via activity of gKir channels) and relaxation, causing vessel dilation. This is suggested to be protective in MCAO.

Many of the steps are well documented, especially those in which KO of TRPA1 in endothelial cells reduces the effects. However, there is a tendency to assume that pharmacological agents will work "as stated on the tin". I have the following issues which need to be resolved before a decision could be made on whether to accept the paper.

1) Choice of vessels to study. The work is done on pial arteries. The authors wish to imply that the dilation produced by TRPA1 in hypoxia will increase blood flow to the brain. To reach this conclusion it would have to be the case that the pial vessels constitute a significant resistance in the vascular pathway between (say) the circle of Willis (where I assume the blood pressure is constant) and the large veins draining the brain. I question whether this is the case, because the pial vessels have multiple anastomoses (to reduce the decrease of brain blood flow if one of them gets blocked: www.ncbi.nlm.nih.gov/pubmed/16379497, www.ncbi.nlm.nih.gov/pubmed/20616030), and so are expected to have a low resistance. Within the brain parenchyma it is known that it is the capillaries that have the highest resistance (www.ncbi.nlm.nih.gov/pubmed/27780904). Can the authors cite evidence that shows that the pial vessels do have a significant resistance?

2) If the resistance argument downplays the significance of pial dilation, is there evidence that can be cited to suggest that downstream penetrating arterioles or contractile capillary pericytes also express TRPA1, and so might increase blood flow by the same mechanism?

3) Use of CPA to block SERCA. I do not understand why SERCA pumps were always blocked with cyclopiazonic acid (CPA) to avoid Ca^2+^ signals from stores, and EGTA-AM was applied. Blocking the SERCA will increase the magnitude of Ca^2+^ transients evoked by TRPA1, since it will reduce removal of Ca^2+^ from the cytoplasm. Lowering the baseline [Ca^2+^] with EGTA will also increase fractional [Ca^2+^] changes. It is therefore essential to carry out at least some experiments showing that 4-HNE evokes detectable Ca^2+^ transients, that are mediated by TRPA1 (i.e. are blocked by A967079 or TRPA1 KO), in endothelial cells that have never been exposed to CPA or to EGTA-AM.

In addition, the Ca^2+^ response of the ECs to the onset of application of CPA should be shown. Presumably [Ca^2+^]cytoplasm rises, but are there then more "spontaneous" transients?

I wonder also whether CPA affects TRPA1 either directly or by elevating [Ca^2+^]i, as thapsigargin has been shown to do (www.ncbi.nlm.nih.gov/pubmed/18515013). If the authors are using CPA instead of thapsigargin because CPA does not affect TRPA1, then they should provide evidence for this.

4) Pharmacology for modulating ROS. The authors' current results disagree with their earlier paper showing a role for NOX2 in generating ROS and 4-HNE (a better explanation of this discrepancy needs to be provided). In this paper they claim that PEG-SOD reduces ROS formation specifically inside the cells. It is essential to add data to Figure 3A showing that there is less ROS generation inside the cells (monitored with DHE) when PEG-SOD was applied. In addition, state for how long the preparation was pre-incubated with the PEG-SOD (and on what time scale it was taken up in previous work) and what pre-incubation time was used for mitoTEMPO.

5) There is an apparent contradiction in Figure 3A (data for mitoTEMPO). The image shows increased DHE fluorescence with hypoxia, while the graph shows a decrease. What is going on?

6) Removal of endothelium. How is it known that passing a bubble through the vessel has no effect on smooth muscle cells (give references and discuss at least)? In Figure 4 why is Tempol used instead of the PEG-SOD used earlier?

7) The preparation. Many experiments are carried out on en face prepared vessels, in which I assume the vessel is cut open so that the endothelial cells can be viewed. It is essential to add a reference to this technique, and a discussion of how this highly invasive procedure may affect Ca^2+^ responses.

(8) Anesthesia. Mice were anesthetized with isoflurane, a putative TRPA1 agonist (www.ncbi.nlm.nih.gov/pubmed/18574153; www.ncbi.nlm.nih.gov/pubmed/28320952) which may affect the overall expression and activation of TRPA1, which is prone to change in the presence of agonists (www.ncbi.nlm.nih.gov/pubmed/25265225). This must be discussed.

(9) Fluo4-AM is said not to enter ECs in subsection “New genetically encoded Ca^2+^ biosensor mice enable optical recording of single-channel TRPA1 activity in the endothelium of intact cerebral arteries.”, yet also in subsection “New genetically encoded Ca^2+^ biosensor mice enable optical recording of single-channel TRPA1 activity in the endothelium of intact cerebral arteries.” EGTA-AM does. What is the explanation of the difference? Is there in fact any evidence that the EGTA gets in (e.g. from monitoring [Ca^2+^]i)?

Reviewer #2:

The authors hypothesize that endothelial TRPA1 can be activated by hypoxia-derived ROS, leading to cerebral artery dilation and reduced stroke injury. The authors begin their examination by showing that 4-HNE and hypoxia increase TRPA1 activity as assessed through sparklet production. Next, they show the rise TRPA1 activity was blocked by antioxidants and TRPA1 antagonists. Moving to myography, hypoxia was revealed to dilate cerebral arteries, a response that was again disrupted by antioxidants, TRPA1 blockade and in eTRPA1-/- mice. Finally, the authors then show that diminishing TRPA1 activity increases stroke volume while TRPA1 activation reduced infarcts in wildtype, but not eTRPA1-/- mice. The authors conclude that endothelial TRPA1 channels are sensitive to hypoxia and that their activation not only leads to vasodilation but can reduce the deleterious effects of ischemic damage.

Overall, this is an interesting study and one that ties the activity of a unique TRP channel to the genesis of stroke injury. Clearly, the strength of the study lies in the cellular measurements of TRPA1 activation. The connection between TRPA1 and stroke injury is perhaps less certain but understandable given the biological scale over which the authors are working. The reviewer outlines three major concerns that require clarification. See below.

1) The first concern is conceptual and centers on the TRPA1 activation by ROS. The authors have previously assigned the production of ROS to NADPH oxidase, yet the current work assigns it to mitochondria. The conceptual issue is this. If TRPA1 is localized in endothelial projections near the smooth muscle contact site, how can ROS generated in mitochondria and outside projections, diffuse to the channel without interacting with other proteins? Is it possible that TRPA1 lies outside the projection in the endothelial body? Clarification in the Discussion section would be warranted.

2) The second concern centers on Figure 5A. The pattern of injury in control mice is not consistent with occlusion of the middle cerebral artery. The injury pattern in the eTRP-/- mice is more in line with control expectations and with that of Figure 5B and 5C. This concern needs to be addressed.

3) The n numbers for the main experiments are low (n=4-5) and should be increased to align with generally accepted practice.

*Reviewer #3:*

Pires and Earley described that TRPA1 channel in endothelial cells of cerebral arteries was activated by hypoxia resulting vasodilation. Hypoxia increased the number of localized Ca^2+^ fluxes through TRPA1 channels called TRPA1 sparklets. These TRPA1 sparklets required mitochondrial ROS but not extracellular O2-. Vasodilation by hypoxia was reduced by blockade or endothelial cell-specific deletion of TRPA1 channels. To investigate functional significance of TRPA1-mediated vasodilation, they assessed the ischemic damage by the middle cerebral artery occlusion (MCAO) and found that the deletion of endothelial TRPA1 channels worsened the damage, while activation of TRPA1 channels reduced the damages, suggesting that importance of TRPA1 activity in cerebral endothelium in ischemic strokes.

This is an interesting paper showing that TRPA1 channels activation during hypoxia may initiate adaptive response to reduce damages by ischemia. More importantly, activation of TRPA1 channels after ischemia may have a beneficial effect on ischemic stroke.

However, there are still major concerns.

1) Authors show that hypoxia produced 4-HNE, an endogenous ligand for TRPA1 channels. However, it is not clear if 4-HNE contributes to TRPA1 sparklets. Based on the findings in Figure 3, TRPA1 channels were activated by mitochondrial ROS. Thus, it seems that 4-HNE do not have a major role in hypoxia-induced TRPA1 sparklets. Please clarify this.

2) Injection of cinnamaaldehyde (CinA) reduced the infarct size in control C57Bl6J mice. It is not clear why they chose this to activate TRPA1. Other TRPA1 activators (4-HNE, AITC) are used in this study and previous literature by the authors. It seems that there is no information that CinA act TRPA1 to dilate cerebral arteries. Thus, it is important to see if CinA increases TRPA1 sparklets in wild type control but not in eTRPA1-/-.

3) MCAO experiments suggest that the deletion of TRPA1 increased infarct size and activation of TRPA1 by CinA reduced infarct size. Authors described that the protective effect of CinA is partially dependent on TRPA1 channels in endothelial cells. However, the authors did not test if CinA fail to reduce infarct size by MCAO in eTRPA1-/-. Also, the authors did not show if CinA reduce infarct size in *TRPA1^fl/fl^* mice. Of course, it may not be appropriate to compare the infarct size between Figure 6A and Figure 6C. But, it is not clear if CinA reduces the infarct size in *TRPA1^fl/fl^*.

4) ROS imaging data is not convincing. It looks DHE fluorescence brighter with hypoxia plus mitoTEMPO. However, there is a trend of decreasing in the summary data.

5) Authors propose that TRPA1-mediated vasodilation improve blood flow and outcomes after ischemic stroke based on the results using endothelium specific TRPA1 channel knockouts. In vitro data suggest that TRPA1 activity cause vasodilation. However, authors did not test if TRPA1-channels activity actually improve blood flow in vivo.

6) CPA was treated in all preparations to detect TRPA1 sparklets. It is not clear if Ca^2+^ flux through TRPA1 channels predominantly contribute to hypoxia-induced vasodilation. The contribution of intracellular Ca^2+^ store should be discussed.

[Editors' note: further revisions were requested prior to acceptance, as described below.]

Thank you for submitting your article "Neuroprotective Effects of TRPA1 Channels in the Cerebral Endothelium Following Ischemic Stroke" for consideration by *eLife*. Your article has been reviewed by three highly regarded peer reviewers and the evaluation has been overseen by David Kleinfeld Reviewing Editor and Richard Aldrich as the Senior Editor.

The reviewers have discussed the reviews with one another and the Reviewing Editor. All agree that the revised manuscript is much improved and has "a certain conceptual appeal". Some sloppiness in the presentation, particularly regarding the quality of the figures, must be corrected prior to a final decision on the manuscript.

Figures:

Missing scale bars Several are far less than compelling and there is a haphazardness to the data presentation. Scale bars are routinely missing, vasomotor responses are variable and of questionable quality, photomicrographs are difficult to interpret, and the calcium data inconsistent.

Figures 1-6. Specify in each legend whether cyclopiazonic acid (CPA) and/or EGTA-AM has been added to the bath.

Figure 1. References to panels in the text need to be corrected. Label axes in times series in panel C.

Figure 2. Label axes in times series in panel B. Improved portioning of the panel sizes would improve the esthetics.

Figure 3. Define the color scales (blue/red) in panel A. Improved portioning of the panel sizes would improve the esthetics.

Figure 4. Improved portioning of the panel sizes would improve the esthetics.

Figure 5. Label axes to times series (vasodilation) in panels A-C.

Figure 6. Label axes to times series in panels A-C.

Figure 7. Add a label for primary antibody in panel A. TRPA1 (red) signals apparently exist outside of Tek (green) signals in panel A. Add a higher magnification image of the overlap as an insert. Note that plotting the logarithm of the intensity is a means to highlight weak but significant signals.

Text

The manuscript includes a number of absolute statement, such as "… spatial spread of hypoxia-evoked TRPA1 sparklets were identical to those of.…", "… NOX inhibitor apocynin had no effect.…", etc. "Identical" and "no" simply do no exist in experimental evidence. Such statements must be presented in statistical terms, such as your use of "… significantly diminished.…" with reference to data with error bars.

---

## [Author Response]

This is a potentially highly important paper that shows how oxidative species lead to vasodilation in pial brain vessels and potentially heightens blood flux during stress, injury, and/or hypoxia. While the reviewers found merit to your work, they also note serious complications. These are the essential issues that must be addressed in the text and/or with new data.Past analysis by the Kleinfeld group and by various computational groups conclude that the bulk of the resistance of the vasculature lies in the capillaries with a smaller fraction in the penetrating arterioles. The current work considers only pial arterioles. At the minimum, it is essential to show that penetrating arterioles contain TRPA1 at densities similar or greater than in the pial arterioles. Ideally, the authors should isolate a pial vessel and show that these vessels also react to activation of TRPA1 by 4-HNE.

We agree that this is an important point, although we also note that older literature maintains that the extracranial and pial arteries provide approximately 50% of overall cerebral vascular resistance(Faraci and Heistad, 1990; Heistad, Marcus and Abboud, 1978). We performed the suggested experiments and the revised manuscript now includes new immunofluorescence images showing that TRPA1 channels are present on endothelial cells within penetrating arterioles (Figure 7). Further, new ex vivo pressure myography experiments demonstrate that 4-HNE, hypoxia, and cinnamaldehyde (CinA) cause dilation of pressurized penetrating arterioles that is blocked by the selective TRPA1 inhibitor A967079 (Figure 7 and Figure 8—figure supplement 2). Taken together, our findings indicate that TRPA1 channels are present and functional in penetrating arterioles and dilate these vessels when activated by chemical agonists and hypoxia. Our data support the possibility that hypoxia-induced activation of TRPA1 causes dilation of pial arteries and penetrating arterioles, which together contribute to cerebral vascular resistance and have an important impact on local hemodynamic control within the brain parenchyma.

The reviewers raised many essential points. All the points listed below need to be addressed. The following are deemed essential and must be addressed with new text or additional experiments.Essential revisions:As noted by reviewer #1"… it is therefore essential to carry out.… experiments showing that 4-HNE evokes detectable Ca^2+^ transients, that are mediated by TRPA1, in endothelial cells that have never been exposed to CPA or to EGTA-AM."

A primary goal of this study was to elucidate the effects of hypoxia on Ca^2+^ influx through TRPA1 channels in the endothelium, recorded as TRPA1 sparklets. CPA was used in our initial Ca^2+^ imaging experiments in order to deplete intracellular Ca^2+^ stores and abolish Ca^2+^ release from the ER, thereby isolating Ca^2+^ influx events. EGTA-AM was used for methodological reasons – in pilot studies, we found that the Ca^2+^ chelator improves signal-to-noise ratio, and Ca^2+^ buffering can theoretically prevent Ca^2+^-dependent channel activation/inactivation. This method of *in situ* visualization of TRP channel sparklets was pioneered by an important study from Dr. Mark Nelson’s laboratory investigating TRPV4 sparklets in mesenteric arteries (Sonkusare et al., 2012). We included discussion of these methodological details in the revised manuscript.

We also see merit in the experiments proposed by the reviewer and have performed the recommended studies. These new data show that 4-HNE significantly increases the frequency of Ca^2+^ transients in the absence of CPA and EGTA-AM, and that the selective TRPA1 inhibitor A967079 blocked this response. An analysis of these new data is presented in Figure 2 and Supplemental file 2 of the revised manuscript.

"There is an apparent contradiction in Figure 3A. The image shows increased DHE fluorescence with hypoxia, while the graph shows a decrease. Please clarify?

We agree that the images of DHE accumulation in endothelial cells after exposure to hypoxia shown in the original manuscript were not representative of the summary data. Image quality has been improved (Figure 4A of the revised manuscript).

Further to Figure 3A, "It is essential to add data.… to demonstrate that there is a reduction of the generation of ROS inside the cells, as monitored with DHE, when PEG-SOD was applied."

We agree and performed new experiments as suggested. We find that incubation of endothelial cells with PEG-SOD (100 U/mL) prevented the increase in DHE fluorescence caused by hypoxia in endothelial cells (Figure 4A of the revised manuscript).

As noted by reviewer #2"[You] previously assigned the production of ROS to NADPH oxidase, yet the current work assigns it to mitochondria..… If TRPA1 is localized in endothelial projections near the smooth muscle contact site, how can ROS generated in mitochondria and outside projections, and diffuse to the channel without interacting with other proteins? Is it possible that TRPA1 lies outside the projection in the endothelial body?" Please clarify.

In our previous study we showed that membrane expression of TRPA1 is enriched within myoendothelial projections and co-localizes with NOX2. We also found that stimulation of NOX2 produced O_2_^-^ that activates the channel through a lipid peroxide product (Sullivan et al., 2017), suggesting that ROS metabolites such as 4-HNE are endogenous activators of TRPA1 in endothelial cells. Interestingly, a recent study from Dr. Donald Welsh’s laboratory showed that mitochondria are localized to the base of myoendothelial projections in endothelial cells (Maarouf et al., 2007). Thus, it is possible that mitochondrial O_2_^-^ can also lead to generation of lipid peroxide metabolites within myoendothelial projections, leading to subsequent activation of TRPA1 channels. We also reported in our previous publication and in the current study (Figure 7) that TRPA1 channels are present on the endothelial cell plasma membrane outside of myoendothelial projections. Thus, TRPA1 channels can potentially be activated by ROS generated through multiple intracellular pathways and by extracellular sources. We discuss this concept in the revised manuscript (Discussion section).

As noted by reviewer # 3"Injection of CinA reduced the infarct size in control.… mice. It is not clear why they chose this to activate TRPA1 [as] there is no information that CinA acts TRPA1 to dilate cerebral arteries. Thus, it is important to see if CinA increases TRPA1 sparklets in wild type control [mice] but not in eTRPA1-/- [mice].

We chose CinA for these studies because this compound has previously been used in vivo at the same dosage used here and was well tolerated by the animals (Huang et al., 2007). To address the reviewer’s concern regarding the effects of CinA on cerebral artery diameter, data presented in the revised manuscript show that CinA dilates pressurized pial arteries and penetrating arterioles in ex vivo pressure myography studies (Figure 8—figure supplement 2). This response is blocked by the TRPA1 inhibitor A967079 and is absent in arteries and arterioles isolated from *Trpa1* ecKO mice. These data demonstrate that CinA dilates pial arteries and penetrating arterioles by activating TRPA1 channels in the endothelium. We did not attempt the suggested Ca^2+^ imaging experiments because the pressure myography studies directly address the reviewer’s concerns.

"[The] authors show that hypoxia produced 4-HNE.… However, it is not clear if 4-HNE contributes to TRPA1 sparklets. Based on the findings in Figure 3, TRPA1 channels were activated by mitochondrial ROS. Thus, it seems that 4-HNE do not have a major role in hypoxia-induced TRPA1 sparklets. Please clarify this issue in the Discussion.

Data shown in Figure 4 show that hypoxia causes generation of O_2_^-^ by mitochondria and data shown in Figure 3 show that hypoxia increases generation of 4-HNE in the endothelium. Prior studies indicate that O_2_^-^ generates lipid peroxide products including 4-HNE (for a review, see Esterbauer, Schaur and Zollner, 1991). Since 4-HNE stimulates TRPA1 sparklets (Figure 1), we believe that it is plausible that hypoxia-induced increases in O_2_^-^ production by mitochondria generates 4-HNE to activate TRPA1 channels at the plasma membrane. We agree that our data do not specifically rule out the possibility that mitochondrial O_2_^-^ directly activates TRPA1 channels. This possibility is discussed in the revised manuscript (Discussion section).

Lastly, please make it clear in the Abstract that you are using isolated vessels for all but the infarct data. " Mouse model" is too nebulous a term.

We have clarified this issue in the Abstract.

Reviewer #1:This paper reports that hypoxia of (pial) arteries leads to the generation of reactive oxygen species such as the lipid peroxide 4-hydroxynonenal (4-HNE), which can activate TRPA1 channels in the endothelial cell surface membrane. These let calcium in, which activates gK(Ca) channels, leading to a K^+^ efflux. This in turn raises [K]o around the arteries' smooth muscle, leading to hyperpolarization (via activity of gKir channels) and relaxation, causing vessel dilation. This is suggested to be protective in MCAO.Many of the steps are well documented, especially those in which KO of TRPA1 in endothelial cells reduces the effects. However, there is a tendency to assume that pharmacological agents will work "as stated on the tin". I have the following issues which need to be resolved before a decision could be made on whether to accept the paper.1) Choice of vessels to study. The work is done on pial arteries. The authors wish to imply that the dilation produced by TRPA1 in hypoxia will increase blood flow to the brain. To reach this conclusion it would have to be the case that the pial vessels constitute a significant resistance in the vascular pathway between (say) the circle of Willis (where I assume the blood pressure is constant) and the large veins draining the brain. I question whether this is the case, because the pial vessels have multiple anastomoses (to reduce the decrease of brain blood flow if one of them gets blocked: www.ncbi.nlm.nih.gov/pubmed/16379497, www.ncbi.nlm.nih.gov/pubmed/20616030), and so are expected to have a low resistance. Within the brain parenchyma it is known that it is the capillaries that have the highest resistance (www.ncbi.nlm.nih.gov/pubmed/27780904). Can the authors cite evidence that shows that the pial vessels do have a significant resistance?

We appreciate the reviewer’s concerns, although we also note that older literature maintains that the extracranial and pial arteries provide approximately 50% of the total vascular resistance in the cerebral circulation (Faraci and Heistad, 1990; Heistad, Marcus and Abboud, 1978). To further address this issue, the revised manuscript includes new data demonstrating that TRPA1 channels are present on the endothelium of penetrating arterioles and that activation of these channels with chemical agonists or hypoxia induced vasodilation. Our data support the possibility that hypoxia-induced activation of TRPA1 causes dilation of pial arteries and penetrating arterioles, which together contribute to cerebral vascular resistance and have an important impact on local hemodynamic control within the brain parenchyma.

2) If the resistance argument downplays the significance of pial dilation, is there evidence that can be cited to suggest that downstream penetrating arterioles or contractile capillary pericytes also express TRPA1, and so might increase blood flow by the same mechanism?

This concern is addressed by our new data showing that activation of TRPA1 with 4-HNE, hypoxia (Figure 7), and CinA (Figure 8—figure supplement 2) induces dilation of pressurized penetrating arterioles, which is blunted by the TRPA1 inhibitor A967079.

3) Use of CPA to block SERCA. I do not understand why SERCA pumps were always blocked with cyclopiazonic acid (CPA) to avoid Ca^2+^ signals from stores, and EGTA-AM was applied. Blocking the SERCA will increase the magnitude of Ca^2+^ transients evoked by TRPA1, since it will reduce removal of Ca^2+^ from the cytoplasm. Lowering the baseline [Ca^2+^] with EGTA will also increase fractional [Ca^2+^] changes. It is therefore essential to carry out at least some experiments showing that 4-HNE evokes detectable Ca^2+^ transients, that are mediated by TRPA1 (i.e. are blocked by A967079 or TRPA1 KO), in endothelial cells that have never been exposed to CPA or to EGTA-AM.In addition, the Ca^2+^ response of the ECs to the onset of application of CPA should be shown. Presumably [Ca^2+^]cytoplasm rises, but are there then more "spontaneous" transients?

A primary goal of this study was to elucidate the effects of hypoxia on Ca^2+^ influx through TRPA1 channels in the endothelium, recorded as TRPA1 sparklets. CPA was used in our initial Ca^2+^ imaging experiments in order to deplete intracellular Ca^2+^ stores and abolish Ca^2+^ release from the ER, thereby isolating Ca^2+^ influx events. EGTA-AM was used for methodological reasons – in pilot studies, we found that the Ca^2+^ chelator improves signal-to-noise ratio, and Ca^2+^ buffering can theoretically prevent Ca^2+^-dependent channel activation/inactivation. This method of *in situ* visualization of TRP channel sparklets was pioneered by an important study from Dr. Mark Nelson’s laboratory investigating TRPV4 sparklets in mesenteric arteries (Sonkusare et al., 2012). We include discussion of these methodological details in the revised manuscript (subsection “New genetically encoded Ca^2+^ biosensor mice enable optical recording of single-channel TRPA1 activity in the endothelium of intact cerebral arteries”).

We also see merit in the experiments proposed by the reviewer and have performed the studies. These new data show that 4-HNE significantly increases the frequency of Ca^2+^ transients in the absence of CPA and EGTA-AM, and that the selective TRPA1 inhibitor A967079 blocked this response. A detailed analysis of these new data is presented in Figure 2 of the revised manuscript.

We did not monitor changes in intracellular Ca^2+^ during initial incubation with CPA in order to minimize bleaching of *GCaMP6f* in the endothelium. We see very little spontaneous Ca^2+^ signaling activity in the unstimulated endothelium after a 15-minute incubation in the presence of CPA (Figure 1), suggesting that the compound does not provoke an increase in spontaneous Ca^2+^ transients. Indeed, we see the opposite and as expected, we see more spontaneous activity in the endothelium of arteries that have not been treated with CPA, likely due to spontaneous Ca^2+^ release from the ER (Figure 2).

I wonder also whether CPA affects TRPA1 either directly or by elevating [Ca^2+^]i, as thapsigargin has been shown to do (www.ncbi.nlm.nih.gov/pubmed/18515013). If the authors are using CPA instead of thapsigargin because CPA does not affect TRPA1, then they should provide evidence for this.

We find no evidence in the literature that CPA directly affects TRPA1 channel activity. Data in the current manuscript show that TRPA1 sparklet frequency is very low in the unstimulated endothelium following treatment with CPA (Figure 1), providing evidence that CPA at the concentration used here has no direct effects on TRPA1 activity. It is also unlikely that an increase in cytosolic [Ca^2+^] caused by CPA affects TRPA1 activity in our experiments because our initial studies were performed in cells treated with EGTA-AM to buffer changes in intracellular [Ca^2+^].

4) Pharmacology for modulating ROS. The authors' current results disagree with their earlier paper showing a role for NOX2 in generating ROS and 4-HNE (a better explanation of this discrepancy needs to be provided). In this paper they claim that PEG-SOD reduces ROS formation specifically inside the cells. It is essential to add data to Figure 3A showing that there is less ROS generation inside the cells (monitored with DHE) when PEG-SOD was applied. In addition, state for how long the preparation was pre-incubated with the PEG-SOD (and on what time scale it was taken up in previous work) and what pre-incubation time was used for mitoTEMPO.

In our previous study we showed that membrane expression of TRPA1 is enriched within myoendothelial projections and co-localizes with NOX2. We also found that stimulation of NOX2 produced O_2_^-^ that activates the channel through a lipid peroxide product metabolite (Sullivan et al., 2015), suggesting that lipid peroxide products such as 4-HNE are endogenous activators of TRPA1 in endothelial cells. Interestingly, a recent study from Dr. Donald Welsh’s laboratory showed that mitochondria are localized to the base of myoendothelial projections in endothelial cells (Maarouf et al., 2007). Thus, it is possible that mitochondrial O_2_^-^ can also lead to generation of lipid peroxide metabolites within myoendothelial projections, leading to subsequent activation of TRPA1 channels. We also reported in our previous publication and in the current study (Figure 7) that TRPA1 channels are also present on the endothelial cell plasma membrane outside of myoendothelial projections. Thus, TRPA1 channels can potentially be activated by ROS generated through multiple intracellular pathways and by extracellular sources. We discuss this concept in the revised manuscript (Discussion section).

We have also included new data (Figure 4A) in the revised manuscript showing that pre-incubation of cerebral arteries with PEG-SOD (100 U/mL) for 30 minutes prior to the onset of hypoxia prevents hypoxia-induced O_2_^-^ generation in endothelial cells. The PEG-SOD concentration and incubation times were selected based on a previous study from the Lombardi laboratory (Lukaszewicz et al., 2016). We have also stated the incubation time for mitoTEMPO (15 minutes) in the methods section of the revised manuscript (subsection “Immunofluorescence labeling”).

5) There is an apparent contradiction in Figure 3A (data for mitoTEMPO). The image shows increased DHE fluorescence with hypoxia, while the graph shows a decrease. What is going on?

We agree that the images of DHE accumulation in endothelial cells after exposure to hypoxia shown in the original manuscript were not representative of the summary data. Image quality has been improved (Figure 4A of the revised manuscript).

6) Removal of endothelium. How is it known that passing a bubble through the vessel has no effect on smooth muscle cells (give references and discuss at least)? In Figure 4 why is Tempol used instead of the PEG-SOD used earlier?

Disruption of the endothelium by passing an air bubble through the lumen is a standard method that was first described in 1989 in a study showing loss of endothelial function without impairing smooth muscle function in skeletal muscle arteries (Ralevic et al., 2012). We cite this study in the revised manuscript. We (and many other laboratories) have used the air bubble technique in many prior studies (for example (Pires et al., 2015; Earley, Gonzales and Crnich, 2009), and did not observe loss of smooth muscle function. All vessels used for the current study maintained their ability to constrict in response to increases in intraluminal pressure following passage of an air bubble through the lumen, indicating that smooth muscle function was intact. We state this in the revised manuscript (subsection “ROS Imaging”).

We used Tempol for these experiments rather than PEG-SOD because in our hands Tempol has better solubility in the physiological bathing solutions used for pressure myography experiments.

7) The preparation. Many experiments are carried out on en face prepared vessels, in which I assume the vessel is cut open so that the endothelial cells can be viewed. It is essential to add a reference to this technique, and a discussion of how this highly invasive procedure may affect Ca^2+^ responses.

Arteries mounted in the *en face* configuration have been used in several previous studies investigating subcellular Ca^2+^ signals in the intact endothelium resulting from release of Ca^2+^ from the ER^12^ and Ca^2+^ influx recorded as TRPV4 sparklets (Sonkusare et al., 2012; Hong et el., 2013; Marziano et al., 2017). Importantly, a study by Sonkusare et al., 2012 showed that TRPV4 sparklets in the *en face* preparation are indistinguishable from those in pressurized arteries, suggesting that longitudinal incision of the artery does not significantly affect the function of Ca^2+^ influx channels in the endothelium. We acknowledge that the endothelium of arteries mounted *en face* is not subjected to the effects of intraluminal pressure or to shear stress due to blood flow, and we cannot study how these stimuli impact Ca^2+^ signaling activity using this preparation. We have included additional references describing the en face preparation and discuss the strengths and limitations of the technique in the revised manuscript (subsection “New genetically encoded Ca^2+^ biosensor mice enable optical recording of single-channel TRPA1 activity in the endothelium of intact cerebral arteries”).

(8) Anesthesia. Mice were anesthetized with isoflurane, a putative TRPA1 agonist (www.ncbi.nlm.nih.gov/pubmed/18574153; www.ncbi.nlm.nih.gov/pubmed/28320952) which may affect the overall expression and activation of TRPA1, which is prone to change in the presence of agonists (www.ncbi.nlm.nih.gov/pubmed/25265225). This must be discussed.

We agree that this is a very interesting point. Prior studies report that isoflurane anesthesia is neuroprotective in various animal models of stroke and subarachnoid hemorrhage (Altay et al., 2012; Altay et al., 2012; Li et al., 2013; Khatibi at al., 2011), but the mechanistic basis of this effect is not known. Our current findings and the studies cited by the reviewer support the concept that isoflurane may provide neuroprotection by potentiating TRPA1-mediated cerebral arterial dilation. This is discussed in the revised manuscript (Discussion section).

(9) Fluo4-AM is said not to enter ECs in subsection “New genetically encoded Ca^2+^ biosensor mice enable optical recording of single-channel TRPA1 activity in the endothelium of intact cerebral arteries.”, yet also in subsection “New genetically encoded Ca^2+^ biosensor mice enable optical recording of single-channel TRPA1 activity in the endothelium of intact cerebral arteries.” EGTA-AM does. What is the explanation of the difference? Is there in fact any evidence that the EGTA gets in (e.g. from monitoring [Ca^2+^]i)?

In pilot studies, we observed that the signal-to-noise ratio of TRPA1 sparklets recorded from *Tek:Gcamp6f* mice is improved by treating the tissue with EGTA-AM, suggesting that it can get into cells. We include this detail in the revised manuscript (subsection “New genetically encoded Ca^2+^ biosensor mice enable optical recording of single-channel TRPA1 activity in the endothelium of intact cerebral arteries”).

Reviewer #2:The authors hypothesize that endothelial TRPA1 can be activated by hypoxia-derived ROS, leading to cerebral artery dilation and reduced stroke injury. The authors begin their examination by showing that 4-HNE and hypoxia increase TRPA1 activity as assessed through sparklet production. Next, they show the rise TRPA1 activity was blocked by antioxidants and TRPA1 antagonists. Moving to myography, hypoxia was revealed to dilate cerebral arteries, a response that was again disrupted by antioxidants, TRPA1 blockade and in eTRPA1-/- mice. Finally, the authors then show that diminishing TRPA1 activity increases stroke volume while TRPA1 activation reduced infarcts in wildtype, but not eTRPA1-/- mice. The authors conclude that endothelial TRPA1 channels are sensitive to hypoxia and that their activation not only leads to vasodilation but can reduce the deleterious effects of ischemic damage.Overall, this is an interesting study and one that ties the activity of a unique TRP channel to the genesis of stroke injury. Clearly, the strength of the study lies in the cellular measurements of TRPA1 activation. The connection between TRPA1 and stroke injury is perhaps less certain but understandable given the biological scale over which the authors are working. The reviewer outlines three major concerns and four minor issues that require clarification. See below.1) The first concern is conceptual and centers on the TRPA1 activation by ROS. The authors have previously assigned the production of ROS to NADPH oxidase, yet the current work assigns it to mitochondria. The conceptual issue is this. If TRPA1 is localized in endothelial projections near the smooth muscle contact site, how can ROS generated in mitochondria and outside projections, diffuse to the channel without interacting with other proteins? Is it possible that TRPA1 lies outside the projection in the endothelial body? Clarification in the Discussion section would be warranted.

In our previous study we showed that membrane expression of TRPA1 is enriched within myoendothelial projections and co-localizes with NOX2. We also found that stimulation of NOX2 produced O_2_^-^ that activates the channel through a lipid peroxide product metabolite (Sullivan et al., 2016), suggesting that lipid peroxide products such as 4-HNE are endogenous activators of TRPA1 in endothelial cells. Interestingly, a recent study from Dr. Donald Welsh’s laboratory showed that mitochondria are localized to the base of myoendothelial projections in endothelial cells^5^. Thus, it is possible that mitochondrial O_2_^-^ can also lead to generation of lipid peroxide metabolites within myoendothelial projections, leading to subsequent activation of TRPA1 channels. We also reported in our previous publication that TRPA1 channels are also present on the endothelial cell plasma membrane outside of myoendothelial projections. Thus, TRPA1 channels can potentially be activated by ROS generated through multiple intracellular pathways and by extracellular sources. We discuss this concept in the revised manuscript (Discussion section).

2) The second concern centers on Figure 5A. The pattern of injury in control mice is not consistent with occlusion of the middle cerebral artery. The injury pattern in the eTRP-/- mice is more in line with control expectations and with that of Figure 5B and 5C. This concern needs to be addressed.

We respectfully disagree with the reviewer, as the cerebral infarct data reported in Figure 8A (previously Figure 6A) for *TRPA1^fl/fl^*mice are not significantly different from the infarct observed in regular C57bl6 mice (Figure 8B, hemisphere infarcted: 44.8 ± 7.1 vs 49.3 ± 4.7, *Trpa1^fl/fl^* vs wildtype C57bl6 mice, p = 0.299). However, the infarct size in *Trpa1* ecKO mice was significantly higher than that from C57bl6 mice (hemisphere infarcted (%): 61.9 ± 3.6, p = 0.036). We have replaced the representative images to better represent the summary data shown in the graphs, as well as to show the similarities and differences between the experiments and genetic backgrounds.

3) The n numbers for the main experiments are low (n=4-5) and should be increased to align with generally accepted practice.

The n values used for the original submission were selected on the basis of power analysis to determine the necessary number of biological replicates to give a generally accepted statistical power of 0.05 for each experiment. However, we acknowledge the reviewer’s point and have increased the *n* values for most of our experiments.

Reviewer #3:Pires and Earley described that TRPA1 channel in endothelial cells of cerebral arteries was activated by hypoxia resulting vasodilation. Hypoxia increased the number of localized Ca^2+^ fluxes through TRPA1 channels called TRPA1 sparklets. These TRPA1 sparklets required mitochondrial ROS but not extracellular O2-. Vasodilation by hypoxia was reduced by blockade or endothelial cell-specific deletion of TRPA1 channels. To investigate functional significance of TRPA1-mediated vasodilation, they assessed the ischemic damage by the middle cerebral artery occlusion (MCAO) and found that the deletion of endothelial TRPA1 channels worsened the damage, while activation of TRPA1 channels reduced the damages, suggesting that importance of TRPA1 activity in cerebral endothelium in ischemic strokes.This is an interesting paper showing that TRPA1 channels activation during hypoxia may initiate adaptive response to reduce damages by ischemia. More importantly, activation of TRPA1 channels after ischemia may have a beneficial effect on ischemic stroke.However, there are still major concerns.1) Authors show that hypoxia produced 4-HNE, an endogenous ligand for TRPA1 channels. However, it is not clear if 4-HNE contributes to TRPA1 sparklets. Based on the findings in Figure 3, TRPA1 channels were activated by mitochondrial ROS. Thus, it seems that 4-HNE do not have a major role in hypoxia-induced TRPA1 sparklets. Please clarify this.

Data shown in Figure 4 show that hypoxia causes generation of mitochondrial O_2_^-^ and data shown in Figure 3 suggest that hypoxia increases generation of 4-HNE in the endothelium. Prior studies indicate that O_2_^-^ can peroxidize membrane lipids to generate lipid peroxide products including 4-HNE (for a review, see Esterbauer, Schaur and Zollner, 1991). Since 4-HNE stimulates TRPA1 sparklets (Figure 1), we believe that it is plausible that hypoxia increases O_2_^-^ production by the mitochondria to generate 4-HNE and activate TRPA1 channels at the plasma membrane. We agree that our data do not specifically rule out the possibility that mitochondrial O_2_^-^ directly activates TRPA1 channel. This possibility is discussed in the revised manuscript (Discussion section).

2) Injection of cinnamaaldehyde (CinA) reduced the infarct size in control C57Bl6J mice. It is not clear why they chose this to activate TRPA1. Other TRPA1 activators (4-HNE, AITC) are used in this study and previous literature by the authors. It seems that there is no information that CinA act TRPA1 to dilate cerebral arteries. Thus, it is important to see if CinA increases TRPA1 sparklets in wild type control but not in eTRPA1-/-.

We chose CinA for these studies because this compound has previously been used in vivo at the same dosage used here and was well tolerated by the animals (Huang et al., 2007). To address the reviewer’s concern regarding the effects of CinA on cerebral artery diameter, data presented in the revised manuscript show that CinA dilates pressurized pial arteries and penetrating arterioles in ex vivo pressure myography studies (Figure 8—figure supplement 2). This response is blocked by the TRPA1 inhibitor A967079 and is absent in arteries and arterioles isolated from *Trpa1* ecKO mice. These data demonstrate that CinA dilates pial arteries and penetrating arterioles by activating TRPA1 channels. We did not attempt the suggested Ca^2+^ imaging experiments because the pressure myography studies directly address the reviewer’s concerns.

3) MCAO experiments suggest that the deletion of TRPA1 increased infarct size and activation of TRPA1 by CinA reduced infarct size. Authors described that the protective effect of CinA is partially dependent on TRPA1 channels in endothelial cells. However, the authors did not test if CinA fail to reduce infarct size by MCAO in eTRPA1-/-. Also, the authors did not show if CinA reduce infarct size in TRPA1^fl/fl^ mice. Of course, it may not be appropriate to compare the infarct size between Figure 6A and Figure 6C. But, it is not clear if CinA reduces the infarct size in TRPA1^fl/fl^.

With respect, we disagree with the reviewer’s comment. In the original submission we tested the effects of CinA injection in the outcome of MCAO in *Trpa1* ecKO mice (Figure 6C in the original submission, Figure 8C in the revised manuscript). These data show that infarct size was smaller in *Trpa1^fl/fl^* (control) mice treated with CinA when compared *Trpa1* ecKO mice treated with CinA.

4) ROS imaging data is not convincing. It looks DHE fluorescence brighter with hypoxia plus mitoTEMPO. However, there is a trend of decreasing in the summary data.

We agree that the images of DHE accumulation in endothelial cells after exposure to hypoxia shown in the original manuscript were not representative of the summary data. Image quality has been improved.

5) Authors propose that TRPA1-mediated vasodilation improve blood flow and outcomes after ischemic stroke based on the results using endothelium specific TRPA1 channel knockouts. In vitro data suggest that TRPA1 activity cause vasodilation. However, authors did not test if TRPA1-channels activity actually improve blood flow in vivo.

The proposed experiments would best be carried out using fMRI or other advanced in vivo imaging modalities. Unfortunately, the capabilities for these potentially elegant studies are not available at our institution. We agree that the suggested experiments would strengthen our conclusions and hope to collaborate with other investigators to more fully characterize the neuroprotective role of TRPA1 channels in the endothelium in an in vivo setting in future studies.

6) CPA was treated in all preparations to detect TRPA1 sparklets. It is not clear if Ca^2+^ flux through TRPA1 channels predominantly contribute to hypoxia-induced vasodilation. The contribution of intracellular Ca^2+^ store should be discussed.

In a prior publication, we showed that the TRPA1 activator AITC stimulated dynamic Ca^2+^ signaling activity in the intact cerebral endothelium that was the result of Ca^2+^ release from IP_3_ receptors on the ER, and that this response was blocked by inhibition of TRPA1 channels (Qian et al., 2013). New data included in the revised manuscript show that 4-HNE increases Ca^2+^ transient activity in the absence of CPA and this response is prevented by the TRPA1 inhibitor A967079. These findings provide evidence that the initial Ca^2+^ signal resulting from Ca^2+^ influx through TRPA1 channels (TRPA1 sparklets) is amplified by Ca^2+^-induced Ca^2+^ release from proximal IP_3_ receptors. We discuss this concept in the revised manuscript (Discussion section).

[Editors' note: further revisions were requested prior to acceptance, as described below.]

The reviewers have discussed the reviews with one another and the Reviewing Editor. All agree that the revised manuscript is much improved and has "a certain conceptual appeal". Some sloppiness in the presentation, particularly regarding the quality of the figures, must be corrected prior to a final decision on the manuscript.

All of the Figures have been improved and some of the photomicrographs have been replaced with higher magnification images.

Figures:Missing scale bars Several are far less than compelling and there is a haphazardness to the data presentation. Scale bars are routinely missing, vasomotor responses are variable and of questionable quality, photomicrographs are difficult to interpret, and the calcium data inconsistent.Figures 1-6. Specify in each legend whether cyclopiazonic acid (CPA) and/or EGTA-AM has been added to the bath.

We have included this information in the legend for each main figure and their supplemental figures.

Figure 1. References to panels in the text need to be corrected. Label axes in times series in panel C.

We corrected references to the figures in the text and we added labels to both the X and Y axes in each of the time series.

Figure 2. Label axes in times series in panel B. Improved portioning of the panel sizes would improve the esthetics.

We added labels to both the X and Y axes in each of the time series plots in panel B and changed the spatial distribution of the panels to improve esthetics.

Figure 3. Define the color scales (blue/red) in panel A. Improved portioning of the panel sizes would improve the esthetics.

The color scale in panel A is now defined in the photomicrograph and in the figure legend (blue: DAPI nuclear staining; red: 4-HNE). We improved the portioning of the panels for better esthetics.

Figure 4. Improved portioning of the panel sizes would improve the esthetics.

We improved the portioning of the panels for better esthetics.

Figure 5. Label axes to times series (vasodilation) in panels A-C.

We added labels to both the X and Y axes in each of the time series in panels A-C. In addition, we have included several new traces of hypoxia-induced vasodilation to illustrate experimental reproducibility.

Figure 6. Label axes to times series in panels A-C.

We added labels to both the X and Y axes in each of the time series in panels A-C. We included new traces of hypoxia-induced vasodilation.

Figure 7. Add a label for primary antibody in panel A. TRPA1 (red) signals apparently exist outside of Tek (green) signals in panel A. Add a higher magnification image of the overlap as an insert. Note that plotting the logarithm of the intensity is a means to highlight weak but significant signals.

We included the label for each primary antibody used in panel A (GFP and TRPA1). The Review Editor points out that TRPA1 seems to be present outside of endothelial cells in our photomicrographs. Astrocytes express TRPA1 channels (Shigetomi et al.,2013; Shigetomi et al., 2011), and it is likely that the signal observed outside of endothelial cells is from TRPA1 channels present in astrocytic end-feet (astrocyte projections that envelop blood vessels within the brain parenchyma). It is important to note that we have not performed co-staining of astrocytes and TRPA1 to confirm this possibility, as this is outside of the scope of the current study. Finally, we have added inserts to the photomicrographs to highlight the overlay between GFP and TRPA1 signals.

TextThe manuscript includes a number of absolute statement, such as "… spatial spread of hypoxia-evoked TRPA1 sparklets were identical to those of.…", "… NOX inhibitor apocynin had no effect.…", etc. "Identical" and "no" simply do no exist in experimental evidence. Such statements must be presented in statistical terms, such as your use of "… significantly diminished.…" with reference to data with error bars.

As per the Review Editor’s suggestion, we now describe our findings in terms of statistical significant differences. “Absolute” terms were used in the initial manuscript to improve readability.